# Is trust a zero-sum game? What happens when institutional sources get it wrong

**Andrew J. Dawson** [ID]*, **Ash Bista, Anne E. Wilson**

Department of Psychology, Wilfrid Laurier University, Waterloo, Canada

* adawson@wlu.ca

## Abstract

Trust in mainstream institutions is declining while people are increasingly turning to alternative media and conspiracy theories. Previous research has suggested that these trends may be linked, but the dynamics of trust across multiple sources has received little investigation. Is trust a neutral process, where each source is judged independently, is it a zero-sum competition, where a loss for one side is a gain for the other, or does losing trust in one source in foster a more generalized sense of distrust? Across three experimental studies (N = 2,951) we examined how people react when a source makes a serious error, testing four potential models of trust dynamics. We found that regardless of whether the outlet is mainstream, counter-mainstream, or neutral, trust drops for the erring source but does not rise for its competitors. This was the case in the context of both food regulations and COVID-19 precautions. Such a pattern suggest that each source may be judged independently of others. However, in several cases, an error made by one source led to a loss of trust in *all* sources, suggesting that rather than choosing sides between competing sources, people are also judging the media landscape as a whole to discern if it is feasible to find trustworthy information. However, correlational data did also find that the more people saw a source as *politicized*, the less they trusted that source and the more they trusted its competitors.

## Introduction

The year 2020 will be forever associated with a crisis that reached every end of the Earth. Within three years, the COVID-19 pandemic claimed over 6 million lives worldwide [1] with over 1 million in the United States alone [2]. A large body of research has shown that several precautions were effective in reducing the spread of the virus, namely wearing facemasks in public places [3–6], getting vaccinated [7–11], and practicing physical distancing [12–14]. Despite the benefits, however, many refused to follow public health guidelines. As late as April 2024, 18.6% of Americans still had not received any dose of the vaccine, and 30.5% had not received the full recommended primary series [15]. In the US and elsewhere, large anti-mask protests were held throughout the pandemic, often during times of surging cases and deaths [16,17]. At the same time as people were disregarding health guidelines, many turned to unfounded conspiracy theories, including that the virus was caused by 5G cell towers [18,19], that there were microchips in the vaccines [20,21], and that the pandemic was planned in order to control the populace [22–24]. Belief in COVID-19 conspiracies persist to this day [25,26].

**Data availability statement:** Data, analysis code, and survey materials are included on the

OSF page for the current research (https://osf.io/hms3q/).

**Funding:** A.W. received an award from the Social Sciences and Humanities Research Council of Canada (#435-2019-1034; https://www.sshrc-crsh.gc.ca/home-accueil-eng.aspx) as well as an award from the Department of National Defence Research Initiative (#877-2019-0007; https://www.sshrc-crsh.gc.ca/funding-financement/programs-programmes/dnd-eng.aspx). A.D. received the Canada Graduate Scholarships – Doctoral (CGS D) from the Social Sciences and Humanities Research Council of Canada (#767-2021-2281; https://www.sshrc-crsh.gc.ca/home-accueil-eng.aspx). No sources of funding played any role in the study design, data collection and analysis, decision to publish, or preparation of the manuscript.

**Competing interests:** The authors have declared that no competing interests exist.

While COVID-19 provided one particularly drastic and consequential example of the fractured informational landscape, broader trends reflecting shifting patterns of trust extend far earlier in time than the pandemic. Trust in public institutions had already been on the decline for since at least the 1970s. For instance, the proportion of Americans reporting a "great deal" of trust in medicine declined by about 30% [27]. A similar trend appears for trust in science, though this may be driven primarily by conservatives [28]. Declining trust has occurred against a backdrop of a number of societal changes, each of which may be related to the phenomenon to some degree, including rising economic inequality [29,30], rising political polarization [31,32], and the emergence of social media and fragmentation of the media environment [33–36]. But what is clear is that the hegemony of traditional institutions is slipping, and this has opened up space for a variety of different sources with very different standards of accuracy. Counter-narratives against the established scientific consensus can be found in many domains such as climate change [37,38], vaccines and autism [39,40] and the origins of HIV [41,42].

What accounts for this dual phenomenon? Why are people turning away from institutional sources that they have traditionally trusted in the past [43,44], and why are some turning towards unreliable sources and even conspiracy theories? Is there a connection between these two trends? These are the questions we intend to investigate in our current studies. We want to know if trust in mainstream institutions is related to trust in these "counter-mainstream" sources, and if so, what exactly is the nature of the relationship. Are these two sides in a zero-sum competition, where a loss in trust for one faction means a gain in trust for the other? Or is the relationship of a different nature? In order to better understand the issue, we turn to several bodies of research, which have examined the subject from different angles.

## Misinformation

One approach has focused on the rise of misinformation (typically defined as claims that lack any evidence or are verifiably false [45–47]). The last decade has seen a high prevalence and dissemination of misinformation throughout the information environment [48–50] making it difficult to verify everything one sees. This is often attributed to the pluralistic and unregulated environment on social media, which is not held to the norms and procedures of traditional news [51]. Indeed, the majority of people now get much of their news from social media, with some using it as their primary source [52,53].

Research on misinformation often focuses on the misinformation on its own, and does not necessarily address the relationship between misinformation and more reliable information. The decision to believe or share misinformation is often viewed as an independent decision, separate from one's decision to believe or share information stemming from mainstream and generally trusted news sources. For example, there is evidence that people fail to fully consider the accuracy of the claims they share online [54,55], and that much of misinformation sharing can be explained in part by individual difference traits such as low cognitive reflection [55,56], reliance on emotion [56], low educational attainment [47], lack of intellectual humility [57], political conservatism [58–60], and overall political extremism [61]. From this perspective, misinformation from unreliable sources can be combatted on its own by encouraging people to be more discerning with the information they encounter online. This is accomplished through accuracy prompts, [55,62–66], fact-checks after headlines [67], and trustworthiness ratings for news sources [68], all of which seem to be at least somewhat effective.

Existing misinformation research, however, does at least indirectly acknowledge the potential connection between belief in misinformation and belief in more reliable sources. Recommendations for interventions emphasize that researchers should not just measure belief

in fake news but belief in real news as well. Interventions that effectively reduce belief in misinformation could also inadvertently reduce belief in *all* information, thereby leading people to dismiss and ignore important facts. The key is to assess discernment between true and false [58,69].

## Trust

Among the research and discourse conducted during and following the COVID-19 pandemic, a common theme has been a lack of trust in traditional institutions. Historically, trust has been placed in certain mainstream institutions, such as scientists and public health agencies. These sources allowed the general public to obtain relevant and accurate information from those with expertise. In the past, trust in such sources has actually been quite high [43,44], which allowed public health messaging to reach its desired audience. But recent years have seen this change. Trust in science and public institutions has been on the decline since the 1970s [27,28], a phenomenon that may have been exacerbated following the COVID-19 pandemic [43,70]. Adherents to the Republican political party in particular are losing trust in the mainstream media amid widespread polarization [71].

At the same time, we have witnessed the rise of alternative media outlets, often producing content with low standards for accuracy [51,72]. These outlets often position themselves directly counter to the mainstream, which they claim is obscuring the truth [73–75]. They typically hold a strong ideological and partisan angle. Although in some periods of history alternative media has been associated more with the left (1960s-1970s), there has recently been a surge in right-wing content in particular [72,74–76]. In many ways, alternative news and its partisan bent marks the widening divide in contemporary western society. The past decade has seen increased polarization on most topics, including issues of science and COVID-19 [77,78], and increased animosity between political factions [31,32]. One major development has been a right-wing backlash against intellectual, government, and corporate institutions that are viewed as culturally at odds with traditional values and the economic interests of common people [79]. These trends have come in tandem with perceptions that traditional news options are highly politicized [80]. Research has shown that perceived politicization of institutions leads to decreased trust, a pattern found regardless of the particular ideology, but that is especially strong when the institution's perceived ideology is opposite that of the observer [81]. Part of reason why alternative media is particularly popular on the right may be because those working in science and the mainstream media tend to be left-leaning [82,83], though some have argued that this is more to do with shift in the Republican party than in the political leaning of the scientists themselves [83].

From this perspective, fact-checking and other efforts to tackle misinformation are addressing only the symptom rather than the disease [84]. Fact-checking may not be effective if people are more inclined to trust the source under scrutiny than the fact-checkers themselves. Accuracy prompts may not be effective for people who believe that alternative sources provide more accurate information than mainstream sources. The problem, then, is not the sudden prevalence of misinformation created by the open internet environment, but the societal divisions that have created diverging visions of reality. Solutions from this viewpoint suggest that misinformation and belief in conspiracy theories are not phenomena that can be considered in isolation. They are intrinsically linked to degrading trust in mainstream institutions. Scholars in this area have focused on how trust in such traditional institutions must be rebuilt, whether that means improving the transparency and neutrality of said institutions [80,81,84] or reducing political polarization [84–86].

Some preliminary research does in fact suggest that trust in these competing sources are closely tied together: the less people trust the mainstream, the more likely they are to trust

alternative news outlets which sometimes includes misinformation and conspiratorial ideas. Trust in mainstream media and government institutions was negatively related to trust in COVID-19 conspiracy theories across multiple correlational [87,88]) and longitudinal [72,87] studies. Although these findings back up the idea of a link between trust in competing news sources, they are limited by potential third variables and limited evidence of causal direction in the context of such a complex, multifaceted issue. Pummerer and colleagues [87] did find that randomly-assigned exposure to a conspiracy theory (that the government and industry was using the pandemic as a cover for data collection) was associated with lower institutional trust. However, this finding may not generalize to any kind of counter-mainstream exposure; the drop in institutional trust may have simply been because the conspiracy theory was explicitly pushing a narrative about why people should not trust the mainstream.

## Potential models of trust

We now turn back to addressing the problem we introduced at the beginning. To summarize, there is considerable evidence that trust in mainstream institutions is on the decline [43,70], at the same time that conspiracies and alternative sources are becoming more prominent [18–21,23,24]. This shift in trust has consequences: The United States has seen over one-million deaths due to COVID-19, despite widespread vaccine availability [2], with a disproportionately high death rate in the most vaccine-skeptical areas [89,90]. In a similar vein, measles, a potentially severe illness once thought to be close to eradication, has experienced a resurgence due to reduced vaccination [91].

However, the nature of the relationship between trust in these opposing sources remains unclear. On some accounts, we have a zero-sum competition where two sides fight over the public's trust. But the evidence is incomplete, and it is unclear whether this is always the case. As researchers, we are interested in better understanding this relationship and its underlying mechanisms. Is trust in opposing sources in fact a zero-sum competition, or can the relationship be better explained with a different framework? In order to answer this research question and set up our empirical investigation, we first consider four potential models of trust that could explain how mainstream and counter-mainstream sources interact in the minds of the public.

**Independent assessment model.** From one perspective, trust might be determined by an assessment of the reliability of a single source of information. In this view, the reliability of each source is presumed to be independent from the reliability of others. If one source becomes more or less trustworthy, the others remain at the same level of trustworthiness as before. For example, if The New York Times publishes something inaccurate, the only effect would be to make The New York Times appear less trustworthy. It would not somehow make Fox News appear more trustworthy. Why would it? They are separate sources of information that can be judged separately. This is what we will call the *Independent Assessment Model*.

As discussed, much of the literature has at least implicitly taken this view by way of omission. Misinformation is treated as a social ill that can be isolated and examined as a separate entity [55,59,62,64–68]. Similarly, trust in mainstream scientific sources is considered on its own without much reference to trust in alternative sources [92–94]. While some initial evidence suggests that competing sources may be more interconnected than is posited by this model [72,87,88], independent assessment remains a distinct possibility that should be given consideration. At the very least, the Independent Assessment Model serves as a sort of null hypothesis against which more complex models can be compared.

**Informational needs model.** Other models of trust dynamics do not assume that trust in each source is evaluated independently but rather propose ways that trust may shift

among sources. Shifts in trust between different news outlets may be a result of an epistemic motivation. For instance, a central component of lay epistemic theory [95] is the need for cognitive closure, reflecting the desire to find definite answers to one's questions. Similarly, news consumption is an information-seeking behaviour driven by people's desire to understand the world around them. For instance, people are often driven to seek out health information [96,97]. Research has shown that people turn to a variety of sources to fulfil their information needs, including not only mainstream news but also social media [98–100]. When trust in mainstream institutions decreases (either due to perceived inaccuracies or for other reasons), people will likely continue to have the same desire for information, but they will be less willing to rely on the mainstream as they did before. This might lead them to turn to alternative sources that they were not as willing to trust previously. Importantly, there is no particular reason why they would turn to one alternate source over the other, so we would predict that the person will increase their trust towards *all* alternate sources as they search for new suppliers of information. These new suppliers will not necessarily be positioning themselves in opposition to the original source. We will call this the *Informational Needs Model.*

**Social alignment model.** The Informational Needs Model assumes that when trust in one source is lost, any alternate source is a viable candidate for information. It does not propose a mechanism for prioritizing alternate sources. However, people may not treat all of their alternate options as equally trustworthy. They may instead rely on heuristics to determine who they should turn to, placing trust in some groups of people more than others. In line with social identity theory [101], humans are predisposed to categorize people into groups based on their different identities. These identities help to determine which groups one is aligned with, which can extend to a heuristic about who can be trusted for information [102,103]. The COVID-19 pandemic provides one prominent example of the importance of social identity in trust. Partisan group membership was one of the strongest predictors of behaviours such as masking, vaccination, and physical distancing, with Democrats more likely to engage in these behaviours than Republicans [104–106]. Democrats are also more likely than Republicans to trust scientific authorities overall [93].

If mainstream and counter-mainstream sources are viewed collectively, as two groups in competition with each other, then trust in either may be perceived as a matter of aligning oneself with one faction against another. Losing trust in one of these two "sides" (e.g., mainstream) may lead people to lean closer and place more trust in the opposing "side" (e.g., counter-mainstream). We will call this the *Social Alignment Model.* In contrast to the informational needs model, the social alignment model predicts that when one source loses credibility, only sources that were posed in opposition to the original source will become more trustworthy. Sources that are not seen as part of the competition (in this case mainstream vs counter-mainstream) will not shift in perceived credibility.

**General loss model.** Originally in developing potential models, we were simply thinking in terms of whether there was a zero-sum competition between sources and of what nature. This led, in our initial theorizing and preregistrations, to the development of our previous three models. Some of the results of our studies, however, suggested a pattern of trust consistent with a fourth potential model that we had not initially identified. This led us to develop one final model.

In the current environment, people have to choose which sources they can trust from a variety of different options. This may lead to a form of competition, where different factions compete for each individual's trust. But alternatively, people may be making broader judgements about the informational landscape in general—whether reliable information is available in the first place, how easily it can be accessed, and how likely they are to successfully find the

truth amidst the chaos of opinion, ambiguity, and misinformation. In such a case, losing trust in any given source (mainstream, counter-mainstream, or anything else) could mean losing trust in *all* sources, as people give up on searching for the truth. We will call this the *General Loss Model.*

Some preliminary evidence suggests that a general "loss of faith" may be occurring in response to the increasingly complex informational environment. Experimental research has shown that exposure to misinformation can decrease trust in general [107], while polling reports that a growing number of people are not only disengaging from the news, but trusting it less in general [108].

### Current research

The current research aimed to investigate whether the dynamic between trust in mainstream and counter-mainstream sources is best characterized as a zero-sum competition or a relationship of a different nature. Specifically, we wanted to test the four competing models of trust that we described above: the *Independent Assessment* Model, the *Informational Needs* Model, the *Social Alignment* Model, and the *General Loss* Model. To do this, we examined shifts in trust following an instance where one of the sources provided inaccurate information. Participants were introduced to a mainstream source, a counter-mainstream source, and a neutral source, and then read a story where one of the three main sources made a serious error in reporting health information.

Naturally, across all models, when a given source makes an error, we expected that specific source to lose trust. But what was most important was how the error would influence trust in the *competing* sources. Each model made unique predictions, which are outlined in our formal hypotheses in Table 1. Hypotheses for each model were preregistered, except for the General Loss Model, which was developed later in the research process. Broadly, the Independent Assessment Model predicted no shifts in trust for any of the other sources. The Informational Needs Model predicted that when one source makes an error and loses trust, *all* other sources will gain trust. The Social Alignment Model predicted that when one source loses trust, only sources that are framed directly in opposition to that source (e.g., counter-mainstream against the mainstream source) will gain trust, while neutral, unaligned sources will not. Finally, the General Loss Model predicted that regardless of which source makes the error, when one source errs, *all* sources will lose trust.

## Materials and methods

All three studies are reported together, as they shared a common rationale and approach. Differences between studies are noted when relevant, but otherwise the basic design was kept the same. All three studies were preregistered. A copy of the preregistrations and surveys for all studies can be found on OSF (https://osf.io/hms3q/). Preregistrations were followed for nearly all analyses. Exploratory analyses are noted. Any divergences from preregistrations are explained in first section of supporting information.

### Participants

Samples consisted of adults living in the United States who self-selected into the studies via CloudResearch MTurk Toolkit [109] and were directed to a survey on Qualtrics [110], provided written informed consent, and were compensated $1.88 USD for participation. In all studies, we recruited equal numbers of Republicans and Democrats based on pre-selection criteria. Ethics approval was obtained from the Wilfrid Laurier University Research Ethics Board (Reference #8302).

**Table 1. Competing hypotheses for each model.**

| **All Models** | |
|---|---|
| Hypothesis 1a | Democrats will trust the primary mainstream source more than Republicans. |
| Hypothesis 1b | Republicans will trust the primary counter-mainstream source more than Democrats. |
| Hypothesis 1c | Trust in the primary neutral source will not differ between Democrats and Republicans. |
| Hypothesis 2a | Trust in the primary mainstream source (FDA) will be lower in the mainstream error condition compared to the control condition. |
| Hypothesis 2b | Trust in the primary counter-mainstream source (National Beacon) will be lower in the counter-mainstream error condition compared to the control condition |
| Hypothesis 2c | Trust in the primary neutral source (Hattrick) will be lower in the neutral error condition compared to the control condition |
| **Independent Assessment Model** | |
| Hypothesis 3a1 | Trust in the primary mainstream source will be the same between the counter-mainstream error and control conditions. |
| Hypothesis 3b1 | Trust in the primary counter-mainstream source will be the same between the mainstream error and control conditions. |
| Hypothesis 4a1 | Trust in the primary neutral source will be the same between the mainstream error, counter-mainstream error, and control conditions. |
| Hypothesis 4b1 | Trust in both the primary mainstream and primary counter-mainstream sources will not differ between the neutral error and control conditions. |
| **Informational Needs Model** | |
| Hypothesis 3a2 | Trust in the primary mainstream source will be higher in the counter-mainstream error condition compared to the control condition. |
| Hypothesis 3b2 | Trust in the primary counter-mainstream source will be higher in the mainstream error condition compared to the control condition. |
| Hypothesis 4a2 | Trust in the primary neutral source will be higher in the mainstream error and counter-mainstream error conditions compared to the control condition. |
| Hypothesis 4b2 | Trust in both the primary mainstream and primary counter-mainstream sources will be higher in the neutral error condition compared to the control condition. |
| **Social Alignment Model** | |
| Hypothesis 3a2 | Trust in the primary mainstream source will be higher in the counter-mainstream error condition compared to the control condition. |
| Hypothesis 3b2 | Trust in the primary counter-mainstream source will be higher in the mainstream error condition compared to the control condition. |
| Hypothesis 4a1 | Trust in the primary neutral source will be the same between the mainstream error, counter-mainstream error, and control conditions. |
| Hypothesis 4b1 | Trust in both the primary mainstream and primary counter-mainstream sources will not differ between the neutral error and control conditions. |
| **General Loss Model** | |
| Hypothesis 3a3 | Trust in the primary mainstream source will be lower in the counter-mainstream error condition compared to the control condition. |
| Hypothesis 3b3 | Trust in the primary counter-mainstream source will be lower in the mainstream error condition compared to the control condition. |
| Hypothesis 4a3 | Trust in the primary neutral source will be lower in the mainstream error and counter-mainstream error conditions compared to the control condition. |
| Hypothesis 4b3 | Trust in both the primary mainstream and primary counter-mainstream sources will be lower in the neutral error condition compared to the control condition. |

Hypotheses 1a-2c are not specific to any one model, while for 3a-4b we consider alternative versions for each model. Identical numbers indicate that the hypothesis is the same between models. Hypotheses 3a & 3b should be used to differentiate the independent assessment model from the informational needs and social alignment models. Hypotheses 4a & 4b should be used to differentiate between the informational needs and social alignment models. The general loss model differs from all other models on Hypotheses 3a-4b.

Desired sample size was determined using statistical power analysis with G*Power [111]. Studies 1 and 2 included four experimental conditions (see procedure below) and we wanted sufficient power to compare each group. According to G*Power, to run multiple comparisons between four groups and detect relatively small effects ($f = .15$ or $\eta_p^2 = .02$) would require 176 per condition or 704 in the final sample. Accounting for an exclusion rate of 15% brings the required initial sample size to 828. Study 3 included only three experimental conditions (see below), but we also wanted to look at the interaction with political party (Democrat vs Republican), meaning comparisons across six groups total, which required a final sample of at 1,056 and an initial sample of 1,242 to account for exclusions.

All three studies included the same set of exclusion criteria to ensure data quality. Participants were excluded if they failed (or did not answer) both of the two attention check items (e.g., "To verify that you are paying attention, please select 1"). An open-ended effort check asked participants what factors they consider when judging the reliability of different sources. Participants were removed if they did not respond or did not discuss anything remotely relevant to the question. An honesty check at the end of the survey asked participants if there was anything wrong with their data. Participants were removed if they indicated that they lied in their responses, chose their answers at random, or accidentally entered incorrect information. Finally, participants were removed if they did not identify as Republican or Democrat in our survey. This last criterion was necessary because we wished to compare across political party as a dichotomous independent factor, and were unable to refer back to CloudResearch recruitment to determine how people initially identified before entering the survey. In Study 3, participants who did not initially identify as Republican or Democrat in the survey were asked a follow-up question about which of the two parties they currently leaned more towards. Participants who indicated a leaning were coded as Republican or Democrat accordingly, and only those who refused to answer the follow-up question were removed from analyses. Only 35 (3%) participants in Study 3 did not select Republican or Democrat in the initial political party measure but did choose one in the follow up question. Results do not show substantial differences when these participants are removed

**Study 1.** An initial sample of 869 was collected on October 4–5, 2022. Applying the exclusion criteria, 21 participants were removed for failing both attention checks, 42 participants failed the effort check, while 45 participants did not identify as Republican or Democrat. No participants were removed based on the honesty check. The final sample included 787 participants (415 Democrat, 372 Republican; 403 women, 382 men, 2 other; 635 White; $M_{Age} = 43.09$, $SD_{Age} = 13.38$).

**Study 2.** An initial sample of 1082 was collected on January 31, 2023. Applying the exclusion criteria, 58 participants were removed for failing both attention checks, 105 participants failed the effort check, 1 participant was removed based on the honesty check, and 60 participants did not identify as Republican or Democrat. The final sample included 942 participants (513 Democrat, 429 Republican; 513 women, 425 men, 2 other; 765 White; $M_{Age} = 44.01$, $SD_{Age} = 12.91$).

**Study 3.** An initial sample of 1287 was collected on August 21–22, 2023. Applying the exclusion criteria, 34 participants were removed for failing both attention checks, 62 participants failed the effort check, while 2 participants did not identify as Republican or Democrat and refused to indicate their preference. No participants were removed based on the honesty check. The final sample included 1222 participants (626 Democrat, 596 Republican; 650 women, 566 men, 6 other; 995 White; $M_{Age} = 43.36$, $SD_{Age} = 12.77$).

## Procedure

**Demographics.** Before the manipulation, participants filled out basic demographic information as well as several measures of their political views. In the current paper we focus on political party. Participants were asked to choose which political party they supported on a categorical measure (Republican, Democrat, Libertarian, Green, Other). In Studies 1 and 2, they were excluded from analyses if they did not choose Republican or Democrat. In Study 3, participants who did not chose Republican or Democrat were asked in a follow up question which of the two they leaned more towards as of the current day. Those that indicated they leaned towards one of the two were coded as supporters of the party they chose, while those who did not answer the follow-up question were excluded.

**Manipulation.** To begin, participants were given a short description of one mainstream source, one counter-mainstream source, and one neutral source. We wanted to make sure that participants were familiar with all three sources regardless of condition, so that trust in each could be measured later in the survey. The exact wording of each description varied by study. See Table S1 in S1 File for a full comparison.

The mainstream source was always a government public health institution. In Study 1 this was the FDA, in Study 2 this was the CDC, and in Study 3 this was the health department of a state in the USA, with the specific state name redacted (ostensibly to avoid bias).

The counter-mainstream source was always a (fictional) fringe news site that positioned itself as skeptical of mainstream sources. The name of the news site was *National Beacon* in Studies 1 and 2. Out of concern that this might sound too much like a specifically right-wing news site, we changed it to *Today News* for Study 3. Today News was described as based in the same state as the health department.

The neutral source was always a (fictional) sports blog called *Hattrick*, which was not meant to have a clear alignment in the mainstream vs counter-mainstream framework. In Study 3, Hattrick was described as based in the same state as the health department.

After reading about the three primary mainstream, counter-mainstream, and neutral sources, participants read a story in which one of those three sources made a serious error in their reporting. The story always consisted of three paragraphs. The first covered how the source initially reported the inaccurate information with a considerable degree of confidence. The reason for the error was kept ambiguous, to avoid bringing any additional factors into play and to align with real-world scenarios where the source's motivations are up for interpretation. The second covered how the source later retracted their previous inaccurate statements (or in the case of Study 2, completely reversed their stance on the issue). The third section discussed how the current data confirms that the source was very inaccurate in their initial statements and that their misreporting may have led to poorer health quality in the United States.

We manipulated which source made the error in the story, either the mainstream source, the counter-mainstream source, or the neutral source. Within each study, the topic and the error were the same regardless of which source reported the incorrect information. All studies also included a control condition, in which participants read an irrelevant story about the Academy Awards, which had nothing to do with health or science topics. The exact wording varied by study and condition. See Table S2 in S1 File for a full comparison.

Different scenarios were used across studies, varying whether the stories were real or fabricated and whether they invoked existing political divisions or represented a relatively novel issue. This approach was taken to provide greater generalizability for our findings and balance experimental control with ecological validity. In Studies 1 and 3, the story was about BNH, a chemical food compound (see above). This was intended to be a relatively neutral topic that would not directly invoke existing political divisions. In Study 2, the story was about the CDC's stance on mask-wearing to protect against COVID-19. Early in the pandemic, the CDC questioned the efficacy of mask wearing, but then changed their stance to recommend it for all Americans [112]. Unlike the BNH scenario, this was a true story, and all information presented consisted of facts participants may have been familiar with. This was intended to be a highly divisive and polarized topic, with clear battle lines and factions [93,105,106,113].

The manipulation in Study 3 differed from Study 1 in several important ways. First, the mainstream source was an unspecified state health department rather than the FDA, and the counter-mainstream source was titled *Today News* rather than *National Beacon*. Second, the neutral error condition was removed in order to lower the required samples size. Third and most importantly, the descriptions of the primary mainstream, counter-mainstream, and

neutral sources were changed to indicate that all three sources were aligned with the participant's political affiliation reported earlier in the survey. (See Table S1 in S1 File.)

This addition to Study 3 was included to account for the possibility that mainstream vs counter-mainstream divisions were confounded with divisions between Republicans and Democrats. We considered that people may be more willing to shift their trust from one source to another when they are assured that all of the sources are aligned with their own politics regardless. We were also interested, however, in whether this would be equally true for Republicans and Democrats. For this reason we recruited a larger sample in order to get sufficient numbers of each party in each condition (see power analyses above).

**Primary trust measures.** Following the manipulation, trust was measured in each primary source (mainstream, counter-mainstream, neutral) in randomized order using the same three-item measure with 7-point scales. Items included "To what degree do you trust [source] as a source of information overall?", "To what degree do you believe that [source] provides accurate information in general?" and "How likely are you to use [source] as a source of information in general?" The [source] mentioned was the specific source in question (e.g., FDA, National Beacon, Hattrick). The measures for the primary mainstream, counter-mainstream, and neutral sources showed good reliability in Study 1, αs =.96,.95,.92, and Study 3, αs =.96,.95,.92.

In Study 2, the original three-item measure of trust was included, as well as a modified version where each item specifically referenced trust in regard to COVID-19 (e.g., "To what degree do you trust [source] as a source of information on COVID-19?"). The general and COVID-specific items showed good reliability together for the mainstream, counter-mainstream and neutral sources, αs =.98,.98,.94, and so were included together.

**Additional trust measures.** As an exploratory measure, we next assessed participants' trust in a variety of additional sources as well, such as the federal government, tabloid magazines, and their local church. Only two items were used to reduce the added length to the survey ("To what degree do you trust each of the following as a source of information overall?" and "How likely are you to **use** each of the following as a source of information in general?"). Participants were told that if they were not familiar with a given source, they should leave the question blank. See Table 2 for full list of additional sources and reliability coefficients. Note that Study 2 also included COVID-specific versions of each item, which were combined with the general items.

**Political influence measures.** In line with past research that highlights the perceptions of politicization as a cause to undermining trust [80,81], 2021), Studies 2 and 3 additionally included single-item measures on a 7-point scale assessing the extent to which the primary sources are influenced by political considerations ("In your personal opinion, how much do you believe that [source] is influenced/motivated by political considerations overall?"). Study 2 only included an item for the primary mainstream source (CDC) while Study 3 included an item on the primary mainstream, counter-mainstream, and neutral sources (state health department, Today News, Hattrick). Notably, in Studies 2 and 3, participants in the control condition were not shown these questions due to a mistake in survey setup.

**Memory checks.** All studies included memory check items at the end of the survey asking participants to recall which source made the error in the story they read, the source's initial stance on the topic (for or against BNH, for or against masks), and whether or not the source changed their stance following the initial inaccurate statements. Notably, this last item had options for whether the source simply retracted their stance (as was the case in Studies 1 and 3) or fully reversed their stance (as was the case in Study 2). The participants were also asked to recall what the data currently indicates regarding BNH (safe, unsafe, neutral, insufficient data) or mask wearing (effective, not effective, insufficient data). These questions were not shown to participants in the control condition.

**Table 2. Additional sources of information with Spearman-Brown reliability coefficients (Studies 1 & 3) and Cronbach's alpha (Study 2).**

| Source | Study 1 | Study 2 | Study 3 |
|---|---|---|---|
| US Federal Government | .89 | .94 | .89 |
| Academic Scientists | .86 | .93 | .87 |
| Washington Post | .90 | .95 | .89 |
| Reuters | .89 | .94 | .90 |
| Wall Street Journal | .88 | .94 | .88 |
| Mainstream media in general | .88 | .94 | .87 |
| Occupy Democrats | .89 | .95 | .89 |
| Breitbart | .91 | .96 | .91 |
| Tabloid Magazines | .88 | .96 | .92 |
| Counter-mainstream media in general | .85 | .92 | .86 |
| Fox News | .95 | .97 | .94 |
| Your local community organizations | .79 | .80 | .80 |
| Sports blogs in general | .81 | .81 | .81 |
| Your local church | .91 | .95 | .92 |
| Your favorite YouTubers | .85 | .93 | .87 |
| Your favorite podcasts | .81 | .92 | .85 |
| Your social media feed | .81 | .93 | .83 |
| Your friends and family | .84 | .93 | .84 |
| Healthcare Professionals/Doctors | — | .92 | .86 |
| Food and Drug Administration (S3 Only) | — | — | .90 |

For Studies 1 & 3, we report reliability for all two-item measures using the Spearman-Brown coefficient, as recommended by Eisinga et al., 2013. Study 2 includes four-item measure with both general and covid-specific items, an as such uses Cronbach's alpha.

## Results

### Memory checks

The majority of participants recalled the correct information in each study respectively for which source made the error (84%, 86%, and 81%) and the source's initial stance on the issue (85%, 86%, 84%). In all studies, a majority also correctly recalled that the source retracted or reversed their stance (81%, 88%, 79%; Across studies, most participants correctly recalled what the data currently indicates about BNH/mask wearing (75%, 81%, 71%). To test the robustness of our results, we ran a version of our main analyses with a sample that only included participants who correctly answered all four of the memory checks. The results are reported in the Tables S3-5 in S1 File in the supporting information. The pattern of results does not substantially differ from the main analyses.

### Main analyses

We ran a mixed ANOVA with trust as the outcome, source (primary mainstream, counter-mainstream, neutral) as a within-subjects predictor, and error condition (mainstream error, counter-mainstream error, neutral error, control) and political party (Republican, Democrat) as between-subjects predictors. Post hoc tests using Fisher's LSD were run for all effects involving more than two groups. As an additional robustness check, the comparisons were also run using a Holm-Bonferonni correction, which did not substantially change the results (see Tables S11-S17 in S1 File). For clarity and ease of comparison across studies, we break

down each main effect and interaction in the model into sections below, with summary and inferential statistics displayed in Tables 3–9.

Our preregistration specified that a between-subjects ANOVA for each trust measure would be used. However, to account for additional within-subject variance, we decided to use a mixed model instead for our main analyses. Separate between-subjects ANOVAs for each of our primary trust variables in each study are reported Tables S6 & S7 in S1 File in the supporting information. The pattern of results for separate between-subjects ANOVAs does not substantially differ from the mixed ANOVAs. It is also possible that when people lose faith in a source, they may not increase their trust in other sources, but they may be more inclined to use them because they are the next best option. To address this concern, we ran a version of our main analyses using only the final item in the trust measure, which asked the participant how likely they were to use the source. The overall pattern of results did not differ from the main analyses. See Tables S8-10 in S1 File for more detail.

**Main effects.** Although our Hypotheses outlined in Table 1 are not written in terms of main effects (having different predictions for the effect of each between-subjects variable on each source), the main effects still provide useful contextual information, which we report here.

Results for the main effect of source are displayed in Table 3. The mainstream source was consistently trusted more than both the counter-mainstream and neutral sources, and the neutral source was trusted more than the counter-mainstream source in Studies 1 and 3.

Results for the main effect of political party are displayed in Table 4. Collapsing across sources, Democrats tended to display more trust than Republicans overall in Studies 2 and 3.

Results for the main effect of error condition are displayed in Table 5. Trust tended to be highest in the control condition, where no error was salient, and lowest in the

**Table 3. Main effect of source, primary trust measures, estimated marginal means and standard errors.**

| | Main Effect | Mainstream Source | Counter-Mainstream Source | Neutral Source |
|---|---|---|---|---|
| Study 1 | $F(2, 1558) = 591.87, \eta_p^2 =.43^{***}$ | 5.00$_a$ (0.05) | 2.92$_b$ (0.05) | 3.31$_c$ (0.05) |
| Study 2 | $F(2, 934) = 696.69, \eta_p^2 =.43^{***}$ | 4.91$_a$ (0.05) | 2.95$_b$ (0.05) | 2.94$_b$ (0.05) |
| Study 3 | $F(2, 2432) = 160.50, \eta_p^2 =.12^{***}$ | 4.59$_a$ (0.04) | 3.79$_b$ (0.04) | 4.09$_c$ (0.04) |

Tables 3–7, 9 and 11 each report different elements of the same mixed ANOVA model. Within each row, means for each condition that do not share a subscript are significantly different. Subscripts do not compare between rows.
†p <.10, * p <.05, ** p <.01, *** p <.001. See Table S11 in S1 File for exact p-values for multiple comparisons.

**Table 4. Main effect of political party, primary trust measures, estimated marginal means and standard errors.**

| | Main Effect | Republican | Democrat |
|---|---|---|---|
| Study 1 | $F(1, 779) = 2.28, \eta_p^2 =.003$ | 3.69$_a$ (0.05) | 3.80$_a$ (0.05) |
| Study 2 | $F(1, 934) = 27.02, \eta_p^2 =.03^{***}$ | 3.43$_a$ (0.05) | 3.77$_b$ (0.04) |
| Study 3 | $F(1, 1216) = 22.52, \eta_p^2 =.02^{***}$ | 4.02$_a$ (0.04) | 4.29$_b$ (0.04) |

Tables 3–7, 9 and 10 each report different elements of the same mixed ANOVA model. Within each row, means for each condition that do not share a subscript are significantly different. Subscripts do not compare between rows.
†p <.10, * p <.05, ** p <.01, *** p <.001

**Table 5. Main effect of error condition, primary trust measures, estimated marginal means and standard errors.**

|  | Main Effect | Mainstream Error | Counter-Mainstream Error | Neutral Error | Control |
|---|---|---|---|---|---|
| Study 1 | $F(3, 779) = 15.27$, $\eta_p^2 = .06$*** | 3.77$_a$ (0.07) | 3.66$_a$ (0.07) | 3.46$_b$ (0.07) | 4.10$_c$ (0.07) |
| Study 2 | $F(3, 934) = 7.68$, $\eta_p^2 = .02$*** | 3.66$_a$ (0.07) | 3.46$_b$ (0.06) | 3.45$_b$ (0.07) | 3.83$_a$ (0.07) |
| Study 3 | $F(2, 1216) = 94.93$, $\eta_p^2 = .14$*** | 3.95$_a$ (0.05) | 3.79$_b$ (0.05) | — | 4.72$_c$ (0.05) |

Tables 3–7, 9 and 10 each report different elements of the same mixed ANOVA model. Within each row, means for each condition that do not share a subscript are significantly different. Subscripts do not compare between rows.
†p <.10, * p <.05 ** p <.01, *** p <.001. See Table S12 in S1 File for exact p-values for multiple comparisons.

counter-mainstream and neutral error conditions, where people's attentions were brought to dubious sources spreading inaccurate information.

**Source × political party (Hypotheses 1a-1c).** The significant interactions between source and political party reveal consistent differences across political groups. Results are displayed in Fig 1 and Table 6. In all three studies, in line with Hypotheses 1a & 1b, Democrats placed more trust in the mainstream source compared to Republicans, while Republicans placed more trust in the counter-mainstream source compared to Democrats. It is worth noting, however, that these are relative differences, and both groups still had higher trust in the mainstream compared to counter-mainstream. As predicted in Hypothesis 1c, we do not see a consistent different between parties for trust in the neutral source, with Republicans significantly higher only in Study 1.

**Source × condition (Hypotheses 2a-4b).** The significant source × error condition interactions reveal how trust in each of the primary sources shifted in response to our manipulation. Results are displayed in Fig 2 and Table 7. Trust in each source is lower in the condition where it makes an error compared to control, which we predicted in Hypotheses 2a-2c. Most important, however, are the findings that distinguish between the four different models of trust we are testing (which make different predictions in Hypotheses 3a-4b). Overall, results appear to be partially consistent with the Independent Assessment Model, where each source is judged on its own without influencing any others. We see no increase in trust relative to control for any source in any condition, which is inconsistent with both of our models that posit a zero-sum competition (Informational Needs and Social Alignment Models).

We do see some support in two of three studies, however, for the General Loss Model, where a loss of trust in one source means a loss of trust for all sources. In Study 1, trust in both the counter-mainstream and neutral sources were lower relative to control when either of the two made an error. In Study 3, trust in all sources was lower than control when any of the sources made an error. Both studies therefore provide evidence that hearing about one source making an error can lead to lower trust in unrelated sources as well. Study 2, however, does not show this pattern, with sources only losing trust relative to control when that specific source in question made the error. Study 2 was the only study to use a real-world issue (COVID-19 masking) which could have activated more pre-established beliefs about source trustworthiness.

To further assess the effects of our manipulation on each trust measure across studies, we ran an internal mini meta-analysis [114]. For ease of interpretation, we focused only on the key comparisons that would directly test the predictions made in Hypotheses 2a-4b, namely the comparisons between the control condition and each of the error conditions. We used

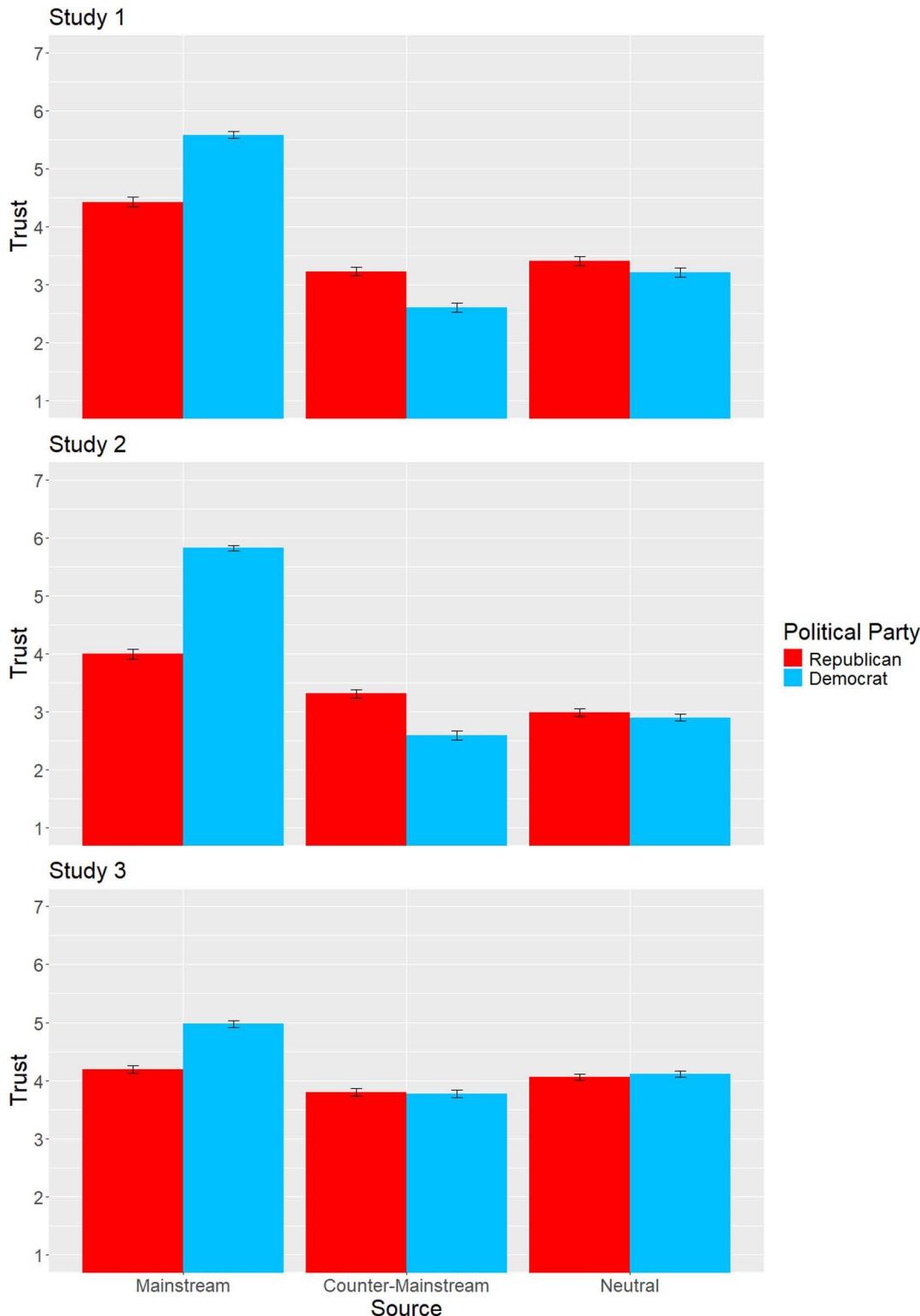

**Fig 1. Mean trust scores with standard errors by political party.**

**Table 6. Source × political party interaction, primary trust measures, marginal means and standard errors (Hypotheses 1a-1c).**

|  | 2-Way Interaction |  | Republican | Democrat |
|---|---|---|---|---|
| Study 1 | $F(2, 1558) = 172.46$, $\eta_p^2 = .12$*** | Mainstream Source | 4.43$_a$ (0.07) | 5.58$_b$ (0.07) |
|  |  | Counter-Mainstream Source | 3.24$_a$ (0.08) | 2.60$_b$ (0.07) |
|  |  | Neutral Source | 3.41$_a$ (0.07) | 3.21$_b$ (0.07) |
| Study 2 | $F(2, 1868) = 238.83$, $\eta_p^2 = .20$*** | Mainstream Source | 3.98$_a$ (0.07) | 5.83$_b$ (0.06) |
|  |  | Counter-Mainstream Source | 3.31$_a$ (0.08) | 2.59$_b$ (0.07) |
|  |  | Neutral Source | 2.99$_a$ (0.07) | 2.89$_a$ (0.06) |
| Study 3 | $F(2, 2432) = 47.51$, $\eta_p^2 = .04$*** | Mainstream Source | 4.20$_a$ (0.06) | 4.99$_b$ (0.06) |
|  |  | Counter-Mainstream Source | 3.79$_a$ (0.06) | 3.78$_a$ (0.06) |
|  |  | Neutral Source | 4.06$_a$ (0.05) | 4.12$_a$ (0.05) |

Tables 3–7, 9 and 10 each report different elements of the same mixed ANOVA model. Within each row, means for each condition that do not share a subscript are significantly different. Subscripts do not compare between rows.
†p <.10, * p <.05 ** p <.01, *** p <.001

the raw means and standard deviations to produce Cohen's *d* values, which were then put into a random-effects model using the Hunter-Schmidt estimator [115,116]. A separate meta-analysis was run for each comparison and each trust measure (mainstream trust, counter-mainstream trust, neutral trust), producing nine overall estimates in total. Note that while these individual meta-analyses are not fully independent from each other (all participants answered all three trust measures), this design does not violate the requirement for independence *within* each meta-analysis [114]. Note also that the meta-analyses for the comparisons between the control and neutral error conditions should be treated with caution, as it only aggregates the effects for Studies 1 and 2 (as Study 3 did not include a neutral error condition).

The results are presented in Table 8. Again, we see that no sources gain trust from the error of another, and the strongest overall effects show that when a given source makes an error, it loses trust. With one exception, all other effects were found to be significant as well. This means that in nearly all cases *all* sources lost trust when *any* source made an error, compared to when no error took place. Such a finding strengthens the support for the General Loss Model, which makes exactly the same prediction. The one nonsignificant effect was for the difference between the control and neutral error conditions for the mainstream source. Though this finding is limited by the fact that only two studies were included in that analysis, it could potentially reflect a somewhat greater robustness of the trust in mainstream sources against the errors of unrelated sources.

**Condition × political party.** There was no significant interaction between error condition and political party in any of the three studies. Results are displayed in Table 9. This interaction was not relevant to any of our hypotheses.

**Three-way interactions.** While there was no significant three-way interaction in Study 1, there was in Studies 2 and 3. To probe these three-way interactions, we examined whether

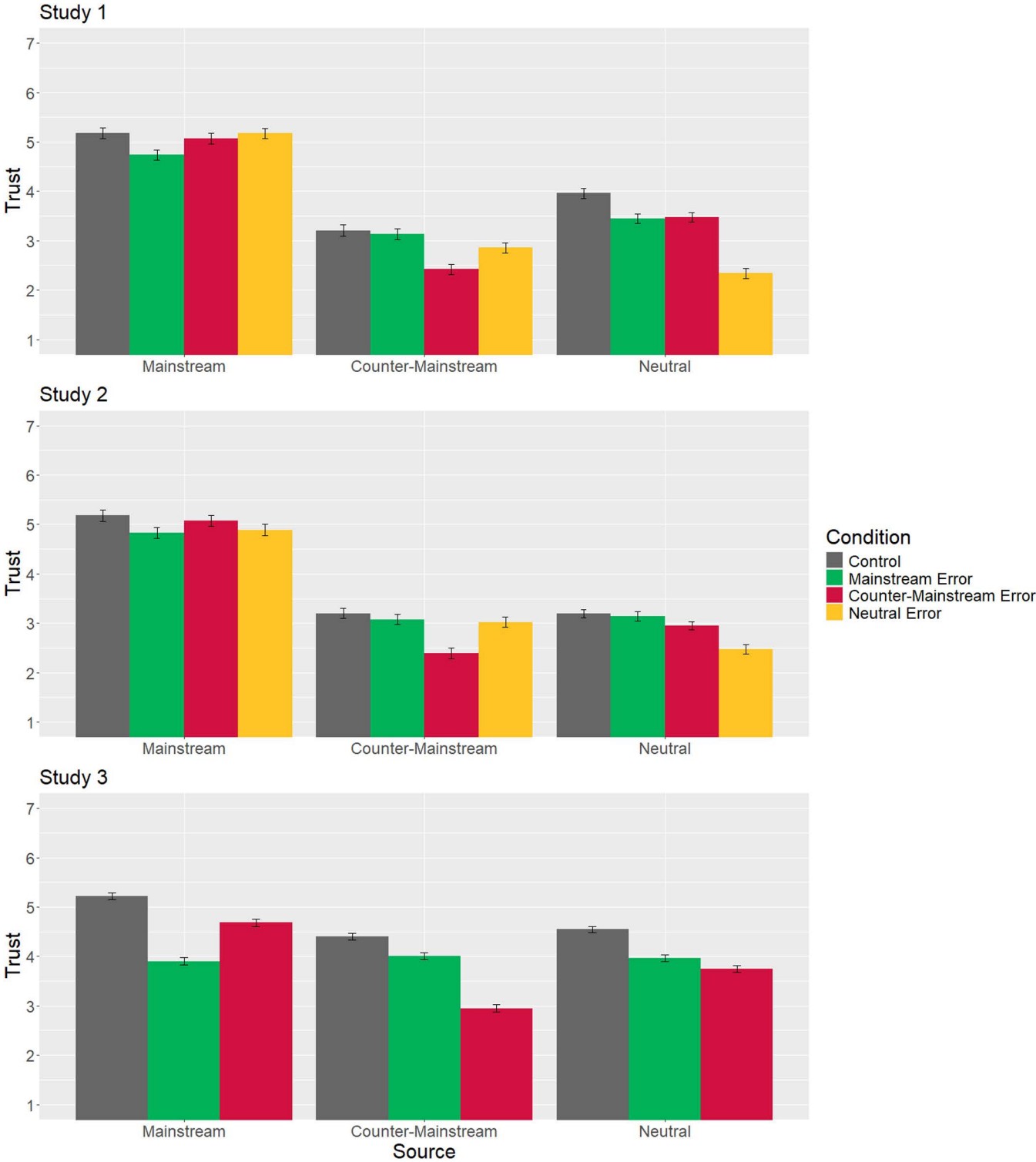

**Fig 2. Mean trust scores with standard errors by experimental condition.**

**Table 7. Source × error condition interaction, primary trust measures, marginal means and standard errors (Hypotheses 2a-4b).**

| | 2-Way Interaction | | Mainstream Error | Counter-Mainstream Error | Neutral Error | Control |
|---|---|---|---|---|---|---|
| Study 1 | $F(6, 1558) = 40.29$, $\eta_p^2 =.09$*** | Mainstream Source | 4.71a (0.10) | 5.07b (0.10) | 5.13b (0.10) | 5.10b (0.10) |
| | | Counter-Mainstream Source | 3.15ac (0.10) | 2.42b (0.10) | 2.88c (0.11) | 3.24a (0.11) |
| | | Neutral Source | 3.45a (0.10) | 3.48a (0.10) | 2.36b (0.10) | 3.95c (0.10) |
| Study 2 | $F(6, 1868) = 10.57$, $\eta_p^2 =.03$*** | Mainstream Source | 4.74a (0.10) | 5.00b (0.09) | 4.83ab (0.10) | 5.08b (0.10) |
| | | Counter-Mainstream Source | 3.11a (0.10) | 2.43b (0.10) | 3.04a (0.10) | 3.22a (0.10) |
| | | Neutral Source | 3.14a (0.09) | 2.95a (0.09) | 2.48b (0.09) | 3.18a (0.09) |
| Study 3 | $F(4, 2432) = 69.29$, $\eta_p^2 =.10$*** | Mainstream Source | 3.90a (0.07) | 4.66b (0.07) | — | 5.22c (0.07) |
| | | Counter-Mainstream Source | 4.00a (0.07) | 2.95b (0.07) | — | 4.40c (0.07) |
| | | Neutral Source | 3.96a (0.07) | 3.75b (0.07) | — | 4.55c (0.07) |

Tables 3–7, 9 and 10 each report different elements of the same mixed ANOVA model. Within each row, means for each condition that do not share a subscript are significantly different. Subscripts do not compare between rows. †p <.10, * p <.05, * p <.01, *** p <.001. See Table S13 in S1 File for exact p-values for multiple comparisons and Table 8 and Figure S1 in S1 File for Cohen's d of key comparisons.

**Table 8. Internal meta-analysis for key comparisons.**

| | | Control — Mainstream Error | | | Control — Counter-Mainstream Error | | | Control — Neutral Error | | |
|---|---|---|---|---|---|---|---|---|---|---|
| **Mainstream Source** | | $n_1$ | $n_2$ | $d$ | $n_1$ | $n_2$ | $d$ | $n_1$ | $n_2$ | $d$ |
| | Study 1 | 196 | 198 | 0.19 | 196 | 197 | 0.04 | 196 | 196 | <.001 |
| | Study 2 | 230 | 239 | 0.13 | 230 | 241 | 0.04 | 230 | 232 | 0.09 |
| | Study 3 | 401 | 413 | 0.63 | 401 | 408 | 0.26 | — | — | — |
| | Estimate | | | 0.48 | | | 0.18 | | | 0.09 |
| | 95% CI | | | [0.10, 0.85] | | | [0.003, 0.36] | | | [-0.04, 0.22] |
| | Z | | | 2.48* | | | 2.00* | | | 1.31 |
| **Counter-Mainstream Source** | | $n_1$ | $n_2$ | $d$ | $n_1$ | $n_2$ | $d$ | $n_1$ | $n_2$ | $d$ |
| | Study 1 | 196 | 198 | 0.03 | 196 | 197 | 0.35 | 196 | 196 | 0.28 |
| | Study 2 | 230 | 239 | 0.05 | 230 | 241 | 0.32 | 230 | 232 | 0.07 |
| | Study 3 | 401 | 413 | 0.23 | 401 | 408 | 0.71 | — | — | — |
| | Estimate | | | 0.16 | | | 0.69 | | | 0.16 |
| | 95% CI | | | [0.02, 0.29] | | | [0.40, 0.97] | | | [0.03, 0.30] |
| | Z | | | 2.32* | | | 4.75*** | | | 2.39* |
| **Neutral Source** | | $n_1$ | $n_2$ | $d$ | $n_1$ | $n_2$ | $d$ | $n_1$ | $n_2$ | $d$ |
| | Study 1 | 196 | 198 | 0.28 | 196 | 197 | 0.26 | 196 | 196 | 0.83 |
| | Study 2 | 230 | 239 | 0.03 | 230 | 241 | 0.15 | 230 | 232 | 0.39 |
| | Study 3 | 401 | 413 | 0.36 | 401 | 408 | 0.46 | — | — | — |
| | Estimate | | | 0.30 | | | 0.39 | | | 0.84 |
| | 95% CI | | | [0.09, 0.50] | | | [0.19, 0.60] | | | [0.41, 1.27] |
| | Z | | | 2.80** | | | 3.75*** | | | 3.80*** |

*Note.* $n_1$ refers to number of participants in control condition, while $n_2$ refers to participants in other condition. Cohen's d values taken from regular means and standard deviations. †p <.10, * p <.05 ** p <.01, *** p <.001

there was a possible condition × political party interaction for each of our sources considered separately (mainstream, counter-mainstream, neutral). Results are displayed in Table 10. Overall, each source typically did not show a simple condition × political party interaction. The simple two-way interaction was only significant for the counter-mainstream source in Study 3, (though it was marginal for the mainstream source in Study 3 and all sources in Study 2). As well, the simple main effects do not show clear differences between political groups. In Study 3, we see the same simple main effects of condition for both Democrats and Republicans: Trust is highest in the control condition, and lowest when the source in question was the one that made the error. In Study 2, the patterns are less clear, though both groups still tended to show lower trust in the condition where the source in question made the error. In both studies, across sources, the main distinction between political parties was that the differences across conditions were more pronounced for Democrats. Overall, however, the three-way interactions do not greatly change the interpretation of our two-way interactions.

## Trust correlations

To better understand the relationship between the trust in each of our primary sources, we examined the correlations between each of the primary trust variables. Results are displayed in Table 11. Notably, in Studies 1 and 3, trust in all three sources were typically positively correlated (with the exception of mainstream and counter-mainstream in Study 1). This is consistent with the General Loss Model, where all sources rise and fall together rather than in competition. It was not the case in Study 2, however, where mainstream and counter-mainstream trust were in fact negatively related (though mainstream and neutral were still positively related). This may reflect the fact that Study 2 examined a real-world issue for which reported trust reflects already-established views. Although we cannot draw strong conclusions based on our correlational results, the finding could indicate that while short-term drops in trust due to a source's error may be most consistent with the General Loss Model, long-term trends may resemble a more oppositional relationship between the mainstream and counter-mainstream, in line with the Social Alignment Model.

**Table 9. Political party × error condition interaction, primary trust measures, marginal means and standard errors.**

| | 2-Way Interaction | | Mainstream Error | Counter-Mainstream Error | Neutral Error | Control |
|---|---|---|---|---|---|---|
| Study 1 | $F(3, 779) = 0.99$, $\eta_p^2 = .004$ | Republican | 3.76$_{ac}$ (0.10) | 3.61$_{ab}$ (0.10) | 3.45$_b$ (0.10) | 3.95$_c$ (0.10) |
| | | Democrat | 3.78$_a$ (0.09) | 3.70$_{ab}$ (0.10) | 3.46$_b$ (0.09) | 4.25$_c$ (0.09) |
| Study 2 | $F(3, 934) = 0.45$, $\eta_p^2 = .001$ | Republican | 3.52$_{ab}$ (0.10) | 3.33$_a$ (0.09) | 3.28$_a$ (0.10) | 3.60$_b$ (0.10) |
| | | Democrat | 3.81$_a$ (0.09) | 3.59$_a$ (0.09) | 3.62$_a$ (0.09) | 4.06$_b$ (0.09) |
| Study 3 | $F(2, 1216) = 1.39$, $\eta_p^2 = .002$ | Republican | 3.77$_a$ (0.07) | 3.72$_a$ (0.07) | — | 4.56$_b$ (0.07) |
| | | Democrat | 4.14$_a$ (0.07) | 3.86$_b$ (0.07) | — | 4.88$_c$ (0.07) |

Tables 3–7, 9 and 10 each report different elements of the same mixed ANOVA model. Within each row, means for each condition that do not share a subscript are significantly different. Subscripts do not compare between rows. †p <.10, * p <.05, ** p <.01, *** p <.001. See Table S14 in S1 File for exact p-values for multiple comparisons.

**Table 10. Source × political party × error condition interaction, primary trust measures, marginal means and standard errors.**

| STUDY 1 | | | | | |
|---|---|---|---|---|---|
| **3-Way Interaction** | | | | | |
| $F(6, 1558) = 0.78$, $\eta_p^2 = .003$ | | | | | |
| **STUDY 2** | | | | | |
| **3-Way Interaction** | | | | | |
| $F(6, 1868) = 3.84$, $\eta_p^2 = .01^{***}$ | | | | | |
| **Mainstream Source** | | | | | |
| Simple 2-Way Interaction | | Mainstream Error | Counter-Mainstream Error | Neutral Error | Control |
| $F(3, 934) = 2.10$, $\eta_p^2 = .01^{\dagger}$ | Republican | 3.98$_{ab}$ (0.14) | 4.02$_{ab}$ (0.14) | 3.76$_a$ (0.14) | 4.23$_b$ (0.14) |
| | Democrat | 5.49$_a$ (0.13) | 5.99$_b$ (0.13) | 5.89$_b$ (0.13) | 5.94$_b$ (0.13) |
| **Counter-Mainstream Source** | | | | | |
| Simple 2-Way Interaction | | Mainstream Error | Counter-Mainstream Error | Neutral Error | Control |
| $F(3, 934) = 2.41$, $\eta_p^2 = .01^{\dagger}$ | Republican | 3.46$_a$ (0.15) | 3.02$_b$ (0.15) | 3.35$_{ab}$ (0.15) | 3.43$_{ab}$ (0.15) |
| | Democrat | 2.77$_a$ (0.14) | 1.85$_b$ (0.14) | 2.73$_a$ (0.14) | 3.01$_a$ (0.14) |
| **Neutral Source** | | | | | |
| Simple 2-Way Interaction | | Mainstream Error | Counter-Mainstream Error | Neutral Error | Control |
| $F(3, 934) = 2.30$, $\eta_p^2 = .01^{\dagger}$ | Republican | 3.11$_a$ (0.13) | 2.97$_{ab}$ (0.13) | 2.73$_b$ (0.13) | 3.14$_a$ (0.14) |
| | Democrat | 3.17$_a$ (0.12) | 2.93$_a$ (0.12) | 2.24$_b$ (0.12) | 3.23$_a$ (0.12) |
| **STUDY 3** | | | | | |
| **3-Way Interaction** | | | | | |
| $F(4, 2432) = 5.12$, $\eta_p^2 = .01^{***}$ | | | | | |
| **Mainstream Source** | | | | | |
| Simple 2-Way Interaction | | Mainstream Error | Counter-Mainstream Error | Neutral Error | Control |
| $F(2, 1216) = 2.41$, $\eta_p^2 = .004^{\dagger}$ | Republican | 3.61$_a$ (0.10) | 4.26$_b$ (0.10) | — | 4.72$_c$ (0.10) |
| | Democrat | 4.18$_a$ (0.10) | 5.06$_b$ (0.10) | — | 5.72$_c$ (0.10) |
| **Counter-Mainstream Source** | | | | | |
| Simple 2-Way Interaction | | Mainstream Error | Counter-Mainstream Error | Neutral Error | Control |
| $F(2, 1216) = 4.47$, $\eta_p^2 = .01^{*}$ | Republican | 3.86$_a$ (0.10) | 3.11$_b$ (0.10) | — | 4.40$_c$ (0.10) |
| | Democrat | 4.14$_a$ (0.10) | 2.80$_b$ (0.10) | — | 4.40$_a$ (0.10) |
| **Neutral Source** | | | | | |
| Simple 2-Way Interaction | | Mainstream Error | Counter-Mainstream Error | Neutral Error | Control |
| $F(2, 1216) = 2.14$, $\eta_p^2 = .004$ | Republican | 3.82$_a$ (0.09) | 3.78$_a$ (0.09) | — | 4.57$_b$ (0.09) |
| | Democrat | 4.10$_a$ (0.09) | 3.72$_b$ (0.09) | — | 4.53$_c$ (0.09) |

Tables 3–7, 9 and 10 each report different elements of the same mixed ANOVA model. Within each row, means for each condition that do not share a subscript are significantly different. Subscripts do not compare between rows. † p <.10, * p <.05, ** p <.01, *** p <.001. See Table S15 in S1 File for exact p-values for multiple comparisons.

**Table 11.** Correlations between trust measures and perceived politicization of sources.

**Study 1**

| | 1 | 2 | 3 |
|---|---|---|---|
| 1 Mainstream Trust | — | — | — |
| 2 Counter−Mainstream Trust | −.07 | — | — |
| 3 Neutral Trust | .07* | .40*** | — |

**Study 2**

| | 1 | 2 | 3 | 4 |
|---|---|---|---|---|
| 1 Mainstream Trust | — | — | — | — |
| 2 Counter−Mainstream Trust | −.16*** | — | — | — |
| 3 Neutral Trust | .01 | .57*** | — | — |
| 4 Mainstream Politicization | −.60*** | .43*** | .26*** | — |

**Study 3**

| | 1 | 2 | 3 | 4 | 5 | 6 |
|---|---|---|---|---|---|---|
| 1 Mainstream Trust | — | — | — | — | — | — |
| 2 Counter−Mainstream Trust | .25*** | — | — | — | — | — |
| 3 Neutral Trust | .32*** | .47*** | — | — | — | — |
| 4 Mainstream Politicization | −.27*** | .18*** | .07† | — | — | — |
| 5 Counter Politicization | .08* | −.14*** | .01 | .33*** | — | — |
| 6 Neutral Politicization | .04 | .09* | −.07* | .40*** | .49*** | — |

†p <.10, * p <.05, ** p <.01, *** p <.001

## Additional sources

As an exploratory analysis, we were also interested in whether people would shift their trust in a number of additional sources beyond the target sources (e.g., federal government, tabloids, local church) in response to our manipulation. To examine this we ran separate ANOVAs for each additional variable, applying a Holm-Bonferroni correction so that each main effect, interaction, and multiple comparison accounted for the number of additional trust variables in the study. See Table S18 in S1 File for full results. Overall, we see that some sources are trusted more by one political party over the other (e.g., Democrats place more trust in federal government and scientists, Republicans in Fox News and church). The experimental manipulation, however, did not show much effect on these variables. In Study 1, "sports blogs in general" were trusted less in the neutral error condition (only) compared to control, while in Study 3, they were trusted less in the counter-mainstream error condition. In Study 3, local community organizations were trusted less in the counter-mainstream error condition. These differences are somewhat consistent with the General Loss Model, where an error in one source leads to decreased trust in unrelated sources. But given that these effects were not consistently found across studies and none of the other exploratory variables showed a difference, we should not place to much weight on them.

What we did find consistently, however, was that in no study, for any of the over 18 additional sources, was trust ever higher in any condition compared to control. This means that for none of the large array of sources, whether mainstream, counter-mainstream, neutral, or otherwise, was there ever any evidence of one source *gaining* trust due to the loss of trust in another source. This rules out a potential alternate scenario that was not accounted for by our main analyses: While the Social Alignment Model assumes that all "mainstream" and "counter-mainstream" sources will be seen as collective entities, two sides united against each other, this may not be the case. Individual mainstream and counter-mainstream sources may simply be

considered in isolation. Conceivably, in this case, if people lose trust in one particular mainstream source (e.g., the FDA), they may not be willing to start trusting the counter-mainstream, but instead simply switch to a *different mainstream source*. However, our additional measures included a number of additional mainstream sources, such as the US federal government, academic scientists, Washington Post, Reuters, and the Wall Street Journal, and yet none gained or lost trust following the error. Similarly, losing trust in the primary counter-mainstream source (e.g., National Beacon) did not lead people to place more trust in additional counter-mainstream outlets such as Occupy Democrats, Breitbart, or tabloid magazines.

## Perceptions of politicization

**Mean differences.** Because past research has found that sources perceived as more politicized are trusted less, we were also interested in how perceptions of politicization differed across condition and political party. Study 2 only measured perceived politicization of the mainstream source, and was examined with a between-subjects ANOVA. Study 3 measured politicization in all three of the mainstream, counter-mainstream, and neutral sources, and was examined with a mixed ANOVA with trust as the outcome, source (primary mainstream, counter-mainstream, neutral) as a within-subjects predictor, and error condition and political party as between-subjects predictors. See Tables 12 and 13 for results.

Overall, the counter-mainstream source was seen as the most politicized, while the neutral source was seen as the least politicized. Perceptions of politicization did not differ across condition, with only one exception: In Study 3, the mainstream source was seen as more politicized in the mainstream error condition than the counter-mainstream error condition. However, because in Studies 2 and 3 participants in the control condition were not asked about perceptions of politicization, we cannot assess how these perceptions compared to baseline. Overall, Republicans tended to perceive more politicization of the mainstream source compared to Democrats, while there were no political differences for the counter-mainstream and neutral source in Study 3. In no studies was there an interaction between the manipulation and political party.

**Table 12. ANOVA for politicization of mainstream source, estimated marginal means and standard errors, Study 2.**

| Error Condition | | | | |
|---|---|---|---|---|
| Main Effect | Mainstream Error | Counter-Mainstream Error | Neutral Error | Control |
| $F(2, 704) = 1.22$, $\eta_p^2 = .003$ | 4.65$_a$ (0.11) | 4.40$_a$ (0.11) | 4.51$_a$ (0.11) | — |

| Political Party | | |
|---|---|---|
| Main Effect | Republican | Democrat |
| $F(1, 704) = 176.40$, $\eta_p^2 = .20^{***}$ | 5.39$_a$ (0.10) | 3.65$_b$ (0.09) |

| Error Condition × Political Party | | | | | |
|---|---|---|---|---|---|
| 2-Way Interaction | | Mainstream Error | Counter-Mainstream Error | Neutral Error | Control |
| $F(2, 704) = 0.59$, $\eta_p^2 = .002$ | Republican | 5.42$_a$ (0.17) | 5.33$_a$ (0.16) | 5.41$_a$ (0.17) | — |
| | Democrat | 3.88$_a$ (0.15) | 3.47$_a$ (0.15) | 3.61$_a$ (0.16) | — |

Within each row, means for each condition that do not share a subscript are significantly different. Subscripts do not compare between rows. †p <.10, * p <.05, ** p <.01 *** p <.001. See Table S16 in S1 File for exact p-values for multiple comparisons.

**Table 13. Mixed ANOVA for politicization measures, estimated marginal means and standard errors, Study 3.**

**Source**

| Main Effect | Mainstream Source | Counter-Mainstream Source | Neutral Source |
|---|---|---|---|
| $F(2, 1620) = 95.30$, $\eta_p^2 = .11$*** | 4.35$_a$ (0.06) | 4.87$_b$ (0.05) | 4.10$_c$ (0.05) |

**Error Condition**

| Main Effect | Mainstream Error | Counter-Mainstream Error | Neutral Error | Control |
|---|---|---|---|---|
| $F(1, 810) = 1.68$, $\eta_p^2 = .002$ | 4.49$_a$ (0.06) | 4.39$_a$ (0.06) | — | — |

**Political Party**

| Main Effect | Republican | | Democrat |
|---|---|---|---|
| $F(1, 810) = 6.59$, $\eta_p^2 = .01$* | 4.55$_a$ (0.06) | | 4.34$_b$ (0.06) |

**Source × Error Condition**

| 2-Way Interaction | | Mainstream Error | Counter-Mainstream Error | Neutral Error | Control |
|---|---|---|---|---|---|
| $F(2, 1620) = 12.33$, $\eta_p^2 = .02$*** | Mainstream | 4.55$_a$ (0.08) | 4.15$_b$ (0.08) | — | — |
| | Counter-Mainstream | 4.79$_a$ (0.07) | 4.95$_a$ (0.07) | — | — |
| | Neutral | 4.15$_a$ (0.07) | 4.06$_a$ (0.07) | — | — |

**Source × Political Party**

| 2-Way Interaction | | Republican | Democrat |
|---|---|---|---|
| $F(2, 1620) = 12.78$, $\eta_p^2 = .02$*** | Mainstream Source | 4.62$_a$ (0.08) | 4.08$_b$ (0.08) |
| | Counter-Mainstream Source | 4.88$_a$ (0.07) | 4.86$_a$ (0.07) |
| | Neutral Source | 4.14$_a$ (0.07) | 4.07$_a$ (0.07) |

**Error Condition × Political Party**

| 2-Way Interaction | | Mainstream Error | Counter-Mainstream Error | Neutral Error | Control |
|---|---|---|---|---|---|
| $F(1, 810) = 0.33$, $\eta_p^2 < .001$ | Republican | 4.62$_a$ (0.08) | 4.47$_a$ (0.08) | — | — |
| | Democrat | 4.37$_a$ (0.08) | 4.31$_a$ (0.08) | — | — |

**Source × Error Condition × Political Party**

| 3-Way Interaction |
|---|
| $F(6, 1620) = 1.45$, $\eta_p^2 = .002$ |

Within each row, means for each condition that do not share a subscript are significantly different. Subscripts do not compare between rows. †p <.10, * p <.05, ** p <.01, *** p <.001. See Table S17 in S1 File for exact p-values for multiple comparisons.

**Correlations.** We were interested in the relationship between perceptions of political influence and trust in each source, given that past research indicates that institutions seen as more politicized are trusted less [80,81]. See Table 11 for correlations between primary

trust measures and perceptions of political influence on each source. (Results did not differ substantially when split by political party. See Tables S19-21 in S1 File for more details.)

In both Studies 2 and 3 we find that when a source was seen as more politicized, it was trusted less. Notably, however, we also found that sources *other* than the target source were trusted *more* when the target source was seen as more politicized. In other words, when the mainstream was seen as more politicized, the counter-mainstream was trusted more, and when the counter-mainstream was seen as more politicized, the mainstream was trusted more. Although we did not observe any experimental evidence for the competition model of trust, these correlational patterns suggest that there may be at least one form of competition at play. If a loss in trust for one source is driven by perceptions of political influence, it could foster a gain in trust for opposing sources. Given that the findings are correlational, however, we are unable to draw causal conclusions. The reverse causal pattern is possible as well, where people simply cast sources they distrust as more politicized, and third variables cannot be ruled out either.

Caution is also warranted in light of the mean partisan differences, namely that Republicans compared to Democrats viewed more politicization for the mainstream source. This is an important reminder that we are not assessing attitudes in a vacuum, and that our participants had pre-existing beliefs going into the survey. It is possible that people are simply more likely to see sources they dislike/do not trust as more politically biased. This alternative causal direction cannot be ruled out based on correlational data.

## Discussion

In light of decreasing trust in public institutions [43,70], increasing prevalence of alternative news media [51,72], and the potential consequences for public health [89,90] researchers have called for more investigation on institutional trust [117]. The current research aimed to contribute to this effort by examining the interplay between trust in competing sources. Despite a growing literature on the importance of institutional trust, there has been relatively little work to date examining the dynamics of trust among different informational sources. When people lose trust in one source, does it have implications for trust in other sources or are all sources judged independently? Are there conditions under which trust may decline for one source and rise for another? More specifically, we wanted to see if this dynamic is best characterized as a zero-sum competition or if trust dynamics are more consistent with another model (represented among our competing hypotheses). While some previous correlational research [72,87,88] has suggested an antagonistic relationship between mainstream and counter-mainstream sources (in line with what we have termed the Informational Needs and Social Alignment Models), it is also possible that each source is judged on its own (Independent Assessment Model) or that inaccuracies from any source will lead to distrust in the informational environment overall (General Loss Model).

### Independent assessment model

Overall, our results indicate that after losing trust in a given source, people do not immediately shift towards the opposing side or search elsewhere for information. This is inconsistent with both of our competition-based models of trust (Informational Needs and Social Alignment). To some extent, our findings offer some support for the Independent Assessment Model, whereby each source of information is judged on its own and does not influence the trust of its competitors. Across all studies we observed a robust tendency for people to reduce their trust in the source that made the error. If trust in informational sources often operates this way, it may be good news. It suggests that when a mainstream institutional source makes

an error, people may lose trust in that source but not immediately shift their trust to other competing sources (a strategy that, on balance, would often result in reliance on lower quality information). Of course, it is also meaningful that people's trust in institutions – especially when the source is a mainstream one – does drop after even a single error. Although trust did not immediately shift to other sources, it is worth recognizing that the relative levels of trust shifted (how much more the mainstream source was trusted relative to other sources) such that the mainstream source was not as prioritized even after a single error. The fragility of trust in these sources underlines the importance of setting a high bar for accuracy and transparency in mainstream sources.

## General loss model

Across studies, we also find evidence for a second pattern: when one source makes an error, people also seem to lose trust across *all* sources, regardless of which outlet in particular made the original error. This is consistent with the General Loss Model. The General Loss Model has important implications for the nature of trust, and speaks to the potentially serious consequences of the deteriorating information environment. If an institutional source makes an error, people may lose trust in that institution, and possibly others. But even if institutions do not misstep, exposure to misinformation from outside sources may have a similar effect, leading people to decrease trust in the news environment at large and give up on trying to stay informed. Given the large amount of misinformation prevalent on the internet [49] and the increased mobility of misinformation in a polarized political environment [85,86] the current situation may pose a particularly strong risk. Indeed, this effect may in part explain trends that we are already seeing, including the long-term decrease in institutional trust [43,70] and increased disengagement and trust in the news [108]. Previous research supports the notion that exposure to misinformation can lower trust across sources, including in mainstream institutions [107,118]. Sometimes this fact is exploited intentionally, as with the "firehose of falsehood" employed by Russian propagandists, in which audiences are overwhelmed by a deluge of misinformation meant to encourage disengagement [119].

## Is trust ever a competition?

Our correlational findings, however, indicate that the immediate effects of our experimental manipulation may only be part of a larger picture. While trust in all of our primary sources were typically positively correlated in Studies 1 and 3, trust in the mainstream and counter-mainstream sources were negatively related in Study 2, which was the only study to examine a real-world, ongoing issue (COVID-19). As well, seeing a source as more politicized was positively associated with trusting its competitors. Though only cross-sectional and correlational, this finding suggests that trust may still act as a zero-sum competition in certain contexts. It is possible that when people lose trust in a source because it made a mistake, this does not incline them towards their competitors. However, when people see a source as more *politically biased*, this could in fact lead people to move to the other side. Future research could examine this possibility experimentally, manipulating the politicization of the source in question to see if this causally increases trust in opposing sources.

An alternative interpretation of our correlation data is that the relationship between people's attitudes extend beyond the immediate effects of our study. In the short term, when told about a serious error in reporting, people may lower their overall trust, raising their guard and becoming more skeptical of anything they hear. In the long run, however, this may not explain how people form or hold opinions towards different sources. Long-term patterns could in fact be characterized by something resembling zero-sum competition, as people much more

gradually shift from trusting one side to trusting another (whether this is due to chronic perceptions of politicization, incompetence, or something else). This would be consistent with previous longitudinal analyses that found that belief in conspiracy theories and orientation to alternative (typically counter-mainstream) news predicts decreased trust in mainstream sources at later timepoints [72,87].

Alternatively, in showing how people see untrusted sources as more politicized, we may have simply highlighted one more way in which people affirm their existing attitudes and allegiances. Indeed, some of the most consistent results across studies were the broad divisions that were likely present from the beginning. In all three studies, both political groups trusted the mainstream source more than the counter-mainstream source overall. At the same time, Democrats placed relatively more trust in the mainstream than Republicans, while Republicans placed relatively more trust in the counter-mainstream than Democrats. The current studies may have captured what happens when people face short-term shocks to the system when hearing about misreporting and misinformation. But the larger determinant of trust is likely still the cultural norms and beliefs of the groups to which people belong.

### Theoretical contributions

The current studies build on the literature on misinformation and institutional trust by providing an in-depth examination of the relationship between trust in opposing sources. Previous research on misinformation has often operated with the implicit assumption that belief in incorrect information can be considered separate from belief in correct information [59,62,64–68]. While some work has looked at interventions that can increase or decrease trust in both accurate and inaccurate information [58,69], it has not closely examined the direct influence they have on each other. Similarly, research on alternative, or what we have labelled "counter-mainstream", sources has often been limited to describing the phenomena on its own [51,72], though some correlational findings do suggest a potential antagonistic relationship between trust in institutional and alternative sources, where higher trust in one is associated with lower trust in the other [72,87,88].

Overall, little work has directly pitted trust in mainstream sources against trust in counter-mainstream sources. The current research did just that in order to discern in greater detail the nature of their relationship. In contrast to some of the early findings [72,87,88], we found that the relationship is not, in fact, consistent with a zero-sum competition, though each source isn't clearly assessed independently either. Rather, trust in these sources may to some degree share a common fate, as people adjust their faith in the media landscape at large.

We have also developed four models to explain the potential dynamics between different sources in the media landscape, namely the Independent Assessment, Informational Needs, Social Alignment, and General Loss Models. While the acute effects of our experimental manipulation suggested support for the Independent Assessment and General Loss Models in this particular context, there may be situations where other models fit better. This possibility is underlined by the correlations between perceptions of politicization and trust, which show a relationship more aligned with our competition-based models (Informational Needs and/or Social Alignment). Future research could use this framework to determine when and where trust is best characterized by a zero-sum competition, and what form that competition takes.

### Limitations and future directions

**Sample.** To the extent that we are primarily interested in the current political context in the United States and similar western democracies, our online samples may be suitable for our purposes. Participants were all living in the United States and English speaking. Both

major political groups were represented roughly equally, as were women and men. The largest limitation in this context was the underrepresentation of racial minorities, with samples overall skewing White. A more representative sample could potentially act differently; while racial minorities are generally liberal and inclined toward the Democratic party [120,121], they may also have historical and contemporary reasons to distrust institutional sources [72] which could lessen the differences between parties that we observed. As well, our choice to focus solely on Republicans and Democrats could limit the applicability of our findings. Using a dichotomous measure of political party did allow us to see clear differences in partisanship and help with ease of interpretation. But this also meant leaving out independents and third parties, who may not clearly fall along the usual battle lines in terms of both political party and the conflict between mainstream and counter-mainstream sources. Finally, we may also be interested in how trust operates outside of this particular political context. In such a case, research should be conducted outside of the United States, ideally in countries with different political and economic structures and less WEIRD cultures [122,123].

**Experimental context.** Participant's attitudes were measured using self-report, which is not always effective in predicting real-world behavior [124]. It is unclear the extent to which our measures captured how people would actually choose to engage with the news and select their sources of information. To actually click on a news story and read it takes more effort than simply clicking on a 1–7 scale. Similarly, stories and headlines online are not typically seen in isolation, but surrounded by additional context, such as likes, comments, and surrounding adjacent headlines. As well, information is not always presented in the relatively neutral fashion employed for our manipulation. Headlines often have a political slant, telling people how to interpret the story, or are recounted second-hand through social media posts or political forums that do not directly link the original story. Sometimes this includes direct messages about who trust; alternative media is known for attacking mainstream sources as unreliable [73,75]. In such cases, additional factors are introduced that may influence people's responses.

Our findings may also be limited by the specific scenarios we used and the informational sources involved. For the sake of simplicity and experimental control, our primary sources included only one mainstream, counter-mainstream, and neutral example in each study. The additional sources included as exploratory measures included an array of mainstream (academic scientists, Washington Post, Reuters, Wall Street Journal) and counter-mainstream (Occupy Democrats, Breitbart, tabloid magazines) sources, as well as those that didn't clearly fit in any category (local church, social media feed, friends and family). But these were not given the same degree of focus, and were not introduced in detail in the same way as the primary sources. Including multiple mainstream and/or counter-mainstream sources in a prominent position could better assess whether people are inclined to switch between options within one "side".

**Politicization.** As discussed above, our correlational findings suggest that trust in different sources may act more like a zero-sum competition in the context of politicization. When people saw a given source (e.g., mainstream) as more politicized, they held greater trust in its competitor (e.g., counter-mainstream). This highlights a key limitation in our manipulation: the intentions of the source were left largely ambiguous, focusing on the error itself rather than any potential political bias. Future research should examine the relationship between perceived politicization and trust experimentally, manipulating the perceived politicization of a given source to see if that influences trust in other, potentially competing, sources. For example, Clark and colleagues [81] manipulated perceptions of politicization vignettes in which a survey conducted on the source's organization found that it was highly biased towards one political ideology or another (left-wing vs right-wing), and that the employees saw this bias as positive. Similar approaches could be used to disentangle the dynamics of trust when one source is experimentally framed as politicized.

**Analytic method.** As discussed, the general loss pattern observed in response to our manipulation may have primarily captured acute effects of misreporting/misinformation (that is, a general loss of trust), and may not have captured long-term trends. People are regularly exposed to large amounts of information, and no single new story is likely to determine the trajectory of their loyalties and beliefs. Future research could use time-lagged analyses see what predicts shifts in trust on a larger time frame. Ultimately though, answering these questions may require us to integrate a variety of research using different methods, in order to gather all the pieces required to understand such a complex phenomenon.

## Conclusion

When faced with an example of misreporting in the media, people's immediate response is not to forgive and forget, nor is it to switch sides. A single error does reliably dampen trust in the offender, but beyond that, when news sources err, people tend to lose trust in the media landscape at large. Even sources that appear to be on opposing sides seem to suffer from the missteps of their competitors, as people grow more skeptical overall. In the current environment, replete with misinformation and political polarization, this finding could have serious implications as people increasingly disengage from the news. As noted by Svoboda [117], experts have recognized how crucial it is to understand the factors leading to institutional distrust and to take evidence-informed steps to mitigate it. This work has contributed a novel method for testing dynamics of trust, offered important insights and identified avenues for future research that could help to address these trends and how to attenuate the risks they present.

## Supporting Information

**S1 File. Is Trust a Zero-Sum Game Supporting Information.**
(DOCX)

## Author contributions

**Conceptualization:** Andrew Dawson, Ash Bista, Anne E. Wilson.

**Data curation:** Andrew Dawson.

**Formal analysis:** Andrew Dawson.

**Funding acquisition:** Anne E. Wilson.

**Investigation:** Andrew Dawson.

**Methodology:** Andrew Dawson, Ash Bista, Anne E. Wilson.

**Project administration:** Andrew Dawson, Anne E. Wilson.

**Resources:** Anne E. Wilson.

**Supervision:** Anne E. Wilson.

**Visualization:** Andrew Dawson.

**Writing – original draft:** Andrew Dawson.

**Writing – review & editing:** Andrew Dawson, Anne E. Wilson.

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
