## [Decision Letter · Decision Letter 0]

10 Oct 2024

PONE-D-24-33343Is trust a zero-sum game? What happens when institutional sources get it wrongPLOS ONE

Dear Dr. Dawson,

Thank you for submitting your manuscript to PLOS ONE. After careful consideration, we feel that it has merit but does not fully meet PLOS ONE’s publication criteria as it currently stands. Therefore, we invite you to submit a revised version of the manuscript that addresses the points raised during the review process.

Both Reviewers raise a number of excellent points. All three of us think that the research questions are interesting and that the methods are well-designed to answer those questions. However, the reviewers raise some important points that should be addressed before publication. These points will strengthen the manuscript, make it more accessible, and allow for stronger conclusions. I will not reiterate all of their points here, but I will draw attention to some of them.

Reviewer 1 makes a number of recommendations to improve the clarity or flow of the paper. Many of these recommendations are worth attending to, and would strengthen the paper. For each, please either implement the change or specify why it would weaken the paper to do so.

Regarding Reviewer 1’s point about the lack of post hoc tests, please have something that gives the specific details of your comparisons. I see that Tables 3-5 & 9 use subscripts to denote which conditions are statistically significant – this is good. However, in the interest of Open Science, it would be good to have more details on these comparisons, such as the effect sizes of the differences, or at least the value of the contrast or post hoc test. (Right now we don’t even know what statistic you used to infer these statistically significant effects.) This information should be put in Supplementary so as to not clog up the main text, except for the nature of the post hoc test (e.g., Tukey, Scheffe) which should be mentioned in the main text.

Please note that I don’t necessarily agree with Reviewer 1’s suggestion about presenting means in text instead of the tables – you have many means and ANOVAs, so in my opinion it makes sense to have them as tables for easy reference. However, do consider whether it makes sense to have *something* in the text – this is especially important for summarizing an effect that is found (or not!) in all three studies, because otherwise that currently requires that readers cross-reference three different tables. Alternately, the three studies could be combined into a single table: in each section (e.g., Source), you could have different lines for each of Study 1, 2, and 3. That would make it easier for readers to see which effects replicate across studies and which effects do not. This comment means that I also don’t necessarily agree with Reviewer 1’s recommendation to present each study separately, as it might actually be easier to present the even more together in one table. However, you should still seriously consider these recommendations and either implement them, or if you think it would weaken the paper to do so, then explain your rationale for not implementing them.

Definitely add error bars to your figures, and specify in the title/caption whether the error bars are SEM, SD, or 95% CI. Error bars are a requirement nowadays at most journals so that readers get a sense of the distribution.

Make sure that your data are publicly available, they’re currently not because it requires permission. You can create an anonymous version of the OSF page, or make your page not require permission to view (make it public after acceptance).

In addition to the Reviewers’ comments, I had the following additional comments of my own:

Major comments:

The results are a little long – they could be trimmed to focus on main findings, and less important results could be relegated to Supplementary (and only referred to in main text).

If the patterns are different in different studies, I strongly recommend an internal meta-analysis (i.e., a meta-analysis of all three studies in the manuscript). This would smooth any chance fluctuations in the effect size or significance, and would allow you to draw a stronger conclusions.

Minor comments:

Table 1 might be easier if it has up & down errors & “no change” for each condition

Tables 3 & 4 might be easier to understand if rearranged or made into more graphs

In Table 3, for the Source x Error interaction, Neutral row, shouldn’t the Counter-Mainstream column have the subscript a instead of b? In general, double check all subscripts

This will not be a light revision, but the manuscript has promise, and in the end I think the revisions will greatly improve the manuscript. We look forward to reading your revision.

We look forward to receiving your revised manuscript.

Kind regards,

Pat Barclay

Academic Editor

PLOS ONE

Journal requirements: When submitting your revision, we need you to address these additional requirements. 1. Please ensure that your manuscript meets PLOS ONE's style requirements, including those for file naming. The PLOS ONE style templates can be found at https://journals.plos.org/plosone/s/file?id=wjVg/PLOSOne_formatting_sample_main_body.pdf and https://journals.plos.org/plosone/s/file?id=ba62/PLOSOne_formatting_sample_title_authors_affiliations.pdf 2. Please provide additional details regarding participant consent. In the ethics statement in the Methods and online submission information, please ensure that you have specified (1) whether consent was informed and (2) what type you obtained (for instance, written or verbal, and if verbal, how it was documented and witnessed). If your study included minors, state whether you obtained consent from parents or guardians. If the need for consent was waived by the ethics committee, please include this information. If you are reporting a retrospective study of medical records or archived samples, please ensure that you have discussed whether all data were fully anonymized before you accessed them and/or whether the IRB or ethics committee waived the requirement for informed consent. If patients provided informed written consent to have data from their medical records used in research, please include this information. 3. Please include captions for your Supporting Information files at the end of your manuscript, and update any in-text citations to match accordingly. Please see our Supporting Information guidelines for more information: http://journals.plos.org/plosone/s/supporting-information. 

Reviewers' comments:

Reviewer's Responses to Questions

**Comments to the Author**

1. Is the manuscript technically sound, and do the data support the conclusions?

Reviewer #1: Partly

Reviewer #2: Yes

2. Has the statistical analysis been performed appropriately and rigorously?

Reviewer #1: I Don't Know

Reviewer #2: Yes

3. Have the authors made all data underlying the findings in their manuscript fully available?

Reviewer #1: No

Reviewer #2: No

4. Is the manuscript presented in an intelligible fashion and written in standard English?

Reviewer #1: Yes

Reviewer #2: Yes

5. Review Comments to the Author

Reviewer #1: The paper explores how individuals react when an informational source makes a serious error, testing various models of trust dynamics. A key finding is that while trust in the erring source decreases, trust in competitors does not correspondingly increase. Given the relevance of misinformation and institutional trust in social and political discourse, the paper’s claims hold potentially significant importance for many fields, including psychology, public health, communitcation, and so on.

The authors have grounded their work in relevant literature surrounding trust, misinformation, and source credibility. However, there are opportunities for improvement in terms of organization and precise language use. Overall, the literature is treated fairly, though the presentation could be clearer.

The data from three experiments provide a potential good foundation for replication, but there are issues with the clarity of the analyses and procedures. The authors report the results of ANOVAs but do not specify whether post hoc tests were conducted to examine group differences. This omission makes it difficult to assess the robustness of the conclusions. Without post hoc analyses (or planned contrasts), it is unclear whether the group differences are statistically significant. The authors should either conduct or clearly report post hoc comparisons to demonstrate that the effects are meaningful. Additionally, improving the organization of the results section would enhance clarity and ease of interpretation.

Regardless of whether the hypotheses were supported or which model was supported (if any), this research demonstrates a good methodological foundation and offers contributions to the field. By examining misinformation, including conspiracy theories, through the lens of trust in institutions, the study examines an essential yet underexplored perspective within the misinformation literature. This approach not only sheds light on the dynamics of how trust influences the acceptance and spread of misinformation but also highlights the critical role institutions play in shaping public perception

Additional comments:

Organization:

• Since the introduction is long, include a brief summary at the beginning to highlight the research question/problem, going straight to the point and outlining what readers should expect from your introduction.

• Avoid short paragraphs and consider merging them (e.g., the first paragraph of the paper).

• Conseder presenting the relevance of the paper more directly. For instance, start with the core issue (e.g., lack of trust in certain institutions) before discussing the consequences of this distrust.

Language Precision:

• Use more precise language (e.g., avoid vague terms like "several"). If possible, quantify or specify where appropriate.

• Ensure that claims are backed by references or examples (e.g., on page 5, the first line of the second paragraph requires examples or sources).

• In sections discussing trust in institutions, include specific numbers to contextualize the severity of the problem.

Materials and Methods:

• Consider moving descriptions of the stories used in the studies from "Current Research" to "Materials" and minimize methodological details in the introduction.

• Ensure the "Procedures" section focuses on what participants did during the study. Avoid repeating information already covered in "Materials."

• Clarify the sequence of events (e.g., whether participants completed demographic questions before or after the manipulation) to improve clarity for replication. Present a step-by-step of what participants did.

Study Presentation:

• Consider reporting each study separately to improve clarity and narrative flow. Present the methods, results, and discussion for Study 1 first. After that, when introducing Study 2 (and later Study 3), emphasize both the similarities and differences between the studies. For example, you could say: "The procedures in Study 2 follow those of Study 1, except that in Study 1 we did X, whereas here we are doing Y." This approach helps maintain continuity while clarifying how each study builds on the previous one. Currently, the presentation is confusing due to the numerous caveats associated with each study.

• Move all measures descriptions (e.g., primary and additional trust measures) into the "Materials" section and do not give any procedures description under "Materials".

• Memory checks do not need to be described in detail. Memory, attention, and manipulation checks are typically mentioned under "Participants" for the whole purpose of informing about exclusions. If they are relevant to your research question, then provide an explanation.

Results and discussion:

• Right after the main analyses heading (paragraph 1), clearly describe the levels of each variable (e.g., error condition, political party) so that readers do not need to refer back to the "Methods" section.

• Clearly spell out the results in the text. While you do this in some sections, other parts are overly technical. Avoid relying too heavily on tables for key findings. Bring in the means and standard errors (SE) directly into the narrative and avoid unnecessary statistical jargon.

• Consider moving the ANOVA tables to supplementary materials, as tables should provide additional details rather than being the primary source of the findings.

• Consider reporting the results for each study separately (Study 1: Methods, Results, Discussion; Study 2: Methods, Results, Discussion, etc.).

• Consider organizing the results in a manner that tells a story. Rename the headings to reflect either the research questions or the key findings, depending on which approach better conveys a cohesive narrative.

• Ensure that graphs include error bars to visually convey variability and precision.

• When you state, "The neutral source was also trusted more than the counter-mainstream source in Studies 1 and 3," did you conduct post hoc tests or planned contrasts to specifically compare between groups? It appears that only ANOVAs were performed, and once you find a significant ANOVA, you just look at the means and infer that group A is greater than group B. However, ANOVAs alone do not indicate which groups differ from one another. Post hoc tests (or planned contrasts) are necessary to draw meaningful conclusions about group comparisons. If the ANOVA results are significant, it only tells you that at least one group mean is different from the others, but it does not reveal which groups differ. ANOVA does not break down these comparisons; it only assesses the overall variance among the groups. For instance, if you have three groups (A, B, and C) and ANOVA finds a significant difference, it doesn't indicate whether A differs from B, A differs from C, or B differs from C. You need post hoc tests for making these sorts of claims. Include these tests to support your interpretations for every group comparison made.

Summary

This paper makes a potentially significant contribution to the field, particularly in the area of trust and misinformation. However, the organizational structure and clarity of the methods and results need refinement for the findings to be fully understood and evaluated.

I could not access the OSF for this project because it required permission.

Reviewer #2: The manuscript “Is Trust a Zero-Sum Game” presents three studies aimed at exploring whether reducing trust in one (e.g. mainstream) source of information has an effect on the trust in other sources, labelled here as counter-mainstream and neutral. To this end, the authors test hypotheses tied to several different models of this dynamic, namely that only the source is affected by its own error (without boosting trust in other sources), that the source is affected which boosts trust in other sources (independent of their alignment), that the source is affected which boosts trust in other sources which are viewed as its “competitors” and then, finally, they add another post-hoc model which describes a situation where trust in all sources decreases after an error of one. I think the set of studies, especially Study 3 which includes political leanings of the participants in the descriptions of the media outlet (very important addition, in my opinion), addresses these questions in a coherent way and I believe these results would be of interest to researchers from several domains.

The manuscript is clearly written and I didn’t have an issue following it at any point (which could have happened, given the number of analyses being reported). The statistical analyses fit the experimental design and I don’t have any comments on these. Unfortunately, I was not able to access the OSF documents with the preregistrations so I didn’t have the chance of comparing what was preregistered with what was done, though this is also mentioned in the supplementary materials, so it didn’t affect my review. (However, this also prevented me from checking whether the data was uploaded. As a note for the future, the authors could make use of the view-only (anonymous) link creation possibility on OSF to provide accessible links during the review process.) On another technical point, I would suggest adding confidence intervals to the graphs in Figures 1 and 2.

On the more theoretical side, there were some issues I’d like to mention that I believe weren’t addressed in the manuscript, but could have been. For one, I was slightly confused with the social alignment model and the rationale behind it. I quote from the ms., “when one source loses trust, only sources that are framed directly in opposition to that source (e.g. counter-mainstream against the mainstream source) will gain trust, while neutral, unaligned sources will not.” I’m not sure I understand this premise, especially when it concerns losing trust in a single counter-mainstream source. I would think that people can just pick another one, given their apparent number in the current informational environment? I would have thought social alignment to have a different effect – that of shifting trust to another counter-mainstream source which provides the individual with the same type of information, but hasn’t shown to have made a large error and thus lost its credibility. Unfortunately, there was no "other from the same category" type in the design.

My general point being, it might be more likely to find zero-sum logic of trust between sources in the same category rather competing ones. We don’t only use information to make decisions that have straightforward consequences on our well-being -- for a lot of the information we do have, it isn’t necessarily strictly important that it’s right as much as that our group might believe it/trust the source. From this view, there could have been another model which would predict that people could increase trust for other similar sources which haven’t lost their credibility. This comment relates to, in part, to Dan Williams’ work on rationalization markets. As far as I understood the design, there wouldn’t be a clear way to tease that apart from the studies as are.

Finally, it isn’t entirely clear from the vignettes how the error correction came about. For example, if it was an intentional error (going against the grain to reach a certain audience) vs. an “honest” mistake which came from lack of initial information, or some other unintentional cause. I think people might view the decision making process behind publishing information by the FDA differently than a media outlet which subsists on clicks. I wonder whether there are different expectations in terms of accuracy tied to the different organizations, which could also affect this dynamic? The gravity of the errors might not be perceived the same either, especially since people probably don’t use e.g. a sports site to learn information on which to make decisions about their health. I think the authors should probably discuss this distinction somewhere in the manuscript.

6. PLOS authors have the option to publish the peer review history of their article (what does this mean? ). If published, this will include your full peer review and any attached files.

**Do you want your identity to be public for this peer review?** For information about this choice, including consent withdrawal, please see our Privacy Policy .

Reviewer #1: No

Reviewer #2: No

---

## [Author Response · Author response to Decision Letter 1]

23 Nov 2024

For full response to reviewer comments, please see attached document labeled "Response to Reviewers November 2024".

---

## [Decision Letter · Decision Letter 1]

28 Jan 2025

PONE-D-24-33343R1Is trust a zero-sum game? What happens when institutional sources get it wrongPLOS ONE

Dear Dr. Dawson,

Thank you for submitting your manuscript to PLOS ONE. After careful consideration, we feel that it is close but does not quite meet PLOS ONE’s publication criteria as it currently stands. Therefore, we invite you to submit a revised version of the manuscript that addresses the points raised during the review process.

 Both reviewers and I think that the revisions have improved the manuscript, and we thank you for your effort in the revision. Reviewer 1 has a few remaining suggestions to improve the manuscript. I have a series of minor corrections that should be made before sending the final revision (see below). My only substantive concern is that Fisher’s LSD post hoc tests are often viewed as too liberal when there are multiple comparisons. You should use something that corrects – even slightly – for multiple comparisons. In the Supplementary Material, it looks like the comparisons are robust, so I’m sure they’d stand up to something more conservative, whether Tukey or possibly even Bonferroni. Readers will be more convinced if you still have significance with a stricter test. And it would be OK to deviate from your pre-registration to say that you’re presenting more conservative tests yet still find a significant effect. I see two ways to deal with this: a) use a post hoc test in main text that at least partially controls for multiple comparisons, describe that you're deviating from your pre-registration to make the analyses more conservative, and present the pre-registered analyses in Supplementary; or b) conduct a post hoc test that at least partially controls for multiple comparisons, keep it in Supplementary, but mention in main text that in addition to your pre-registered Fisher's LSD you also did a robustness check by doing an additional analysis to control for multiple comparisons, and it produced the same results (assuming that it does so). I lean towards the second solution - it keeps the focus on the pre-registered analysis, but shows that it's also robust to more conservative analyses, without bogging down the main analyses too much. Here are a number of very minor suggestions:

Line 68 pluralize “vaccine”Lines 424-425 “academy awards” should be capitalized as Academy AwardsIn Tables, where saying that subscripts apply to the row, perhaps clarify by explicitly saying that they *don’t* apply between rowsLine 724: Under “Trust correlations”, trust in all three sources was *not* typically positively correlated – they’re sometimes negatively correlated. Please correct how this is describedTable 11 Study 3: Presumably “Counter” means “Counter-Mainstream”. Can this be spelled out in full?Line 776: Missing “differed”?Table 13: in the rows with Source, perhaps specify Mainstream “source”, Counter-Mainstream “source”, and Neutral “source” (i.e., add “source” to the text to help readers figure out quicker what’s varying in rows vs. in columns)

We look forward to receiving your revised manuscript.

Kind regards,

Pat Barclay

Academic Editor

PLOS ONE

Journal Requirements:

Reviewers' comments:

Reviewer's Responses to Questions

**Comments to the Author**

1. If the authors have adequately addressed your comments raised in a previous round of review and you feel that this manuscript is now acceptable for publication, you may indicate that here to bypass the “Comments to the Author” section, enter your conflict of interest statement in the “Confidential to Editor” section, and submit your "Accept" recommendation.

Reviewer #1: All comments have been addressed

Reviewer #2: All comments have been addressed

2. Is the manuscript technically sound, and do the data support the conclusions?

Reviewer #1: Yes

Reviewer #2: Yes

3. Has the statistical analysis been performed appropriately and rigorously?

Reviewer #1: Yes

Reviewer #2: Yes

4. Have the authors made all data underlying the findings in their manuscript fully available?

Reviewer #1: Yes

Reviewer #2: Yes

5. Is the manuscript presented in an intelligible fashion and written in standard English?

Reviewer #1: Yes

Reviewer #2: Yes

6. Review Comments to the Author

Reviewer #1: Thank you for addressing my previous concerns. The manuscript is much clearer now, and I was able to access the OSF page without any issues.

The methods and procedures are also much clearer in this revision, which strengthens the overall presentation. I also found the results section to be cohesive and easy to follow, which enhances the clarity of the findings.

Additionally, while you clearly explain that the hypotheses related to the general loss model were not pre-registered, this distinction should be explicitly marked in the hypothesis table itself. Doing so will ensure readers are immediately aware (or reminded) of which hypotheses were pre-registered and which were not, helping to maintain clarity throughout.

Also, I have just one minor suggestion regarding the numbering of hypotheses. I understood what they referred to after reading the captions in your hypothesis table. However, the numbering can be a bit confusing at first glance—especially since seeing 1, 2, and 3 naturally leads readers to associate them with the studies. This might be something to consider for added clarity.

Aside from these minor points, I have no further suggestions. I believe the paper is in good shape for publication.

Best regards, Reviewer 1.

Reviewer #2: I am very happy to have received this manuscript for re-revision in its new form, which I believe has greatly improved. I'm glad the authors added clarifications on the comments I made regarding theory (and am satisfied with their replies), but I also think that the presentation of the data now flows more smoothly than in the previous version. I've re-checked the osf link for the project which is now accessible, and I appreciate the redone figures with the error bars. All in all, I don't have any other suggestions for improvement at this point.

7. PLOS authors have the option to publish the peer review history of their article (what does this mean? ). If published, this will include your full peer review and any attached files.

**Do you want your identity to be public for this peer review?** For information about this choice, including consent withdrawal, please see our Privacy Policy .

Reviewer #1: No

Reviewer #2: No

---

## [Author Response · Author response to Decision Letter 2]

7 Mar 2025

[Please refer to "Response to Reviewers March 2025" document for a better-formatted version of the response below.]

Dear Dr. Barclay and reviewers,

Thank you for your further comments. Your additional concerns were helpful and we believe the further revisions make for a stronger paper. We include all comments below followed by our response to each point.

My only substantive concern is that Fisher’s LSD post hoc tests are often viewed as too liberal when there are multiple comparisons. You should use something that corrects – even slightly – for multiple comparisons. In the Supplementary Material, it looks like the comparisons are robust, so I’m sure they’d stand up to something more conservative, whether Tukey or possibly even Bonferroni. Readers will be more convinced if you still have significance with a stricter test. And it would be OK to deviate from your pre-registration to say that you’re presenting more conservative tests yet still find a significant effect. I see two ways to deal with this: a) use a post hoc test in main text that at least partially controls for multiple comparisons, describe that you're deviating from your pre-registration to make the analyses more conservative, and present the pre-registered analyses in Supplementary; or b) conduct a post hoc test that at least partially controls for multiple comparisons, keep it in Supplementary, but mention in main text that in addition to your pre-registered Fisher's LSD you also did a robustness check by doing an additional analysis to control for multiple comparisons, and it produced the same results (assuming that it does so). I lean towards the second solution - it keeps the focus on the pre-registered analysis, but shows that it's also robust to more conservative analyses, without bogging down the main analyses too much.

The suggestion to report a version of the analyses using multiple corrections in the supporting information is a good idea. It allows us to test the robustness of our results without complicating things in the main text regarding the preregistration. Since Supplemental Tables S11-S17 already report the detailed p-values for all multiple comparisons, we decided to simply add asterisks to all values that remained significant after applying a Holm-Bonferonni correction. We decided to use the Holm-Bonferonni method for the sake of consistency as we had already used it as well in Table S18, where we report the results for the additional sources.

The Holm-Bonferroni method orders all p-values in the set from lowest to highest. The first p-value is considered significant if it is smaller than the critical value (in this case α = .050) divided by the number of comparisons (k). The second p-value is significant if it is smaller than α/(k-1), the third if it is smaller than α/(k-2), and so on until stopping at the first comparison to be found as non-significant. For example, for each trust variable in Table S13, we have 6 comparisons. So the smallest p-value had to be smaller than .050/6 = .008, the second smallest had to be smaller than .050/5 = .010, the third .050/4 = .013, and so on.

The results were largely the same with the Holm-Bonferroni correction, with nearly all comparisons relevant to our hypotheses remaining significant. We have added a sentence in the main text to indicate this decision, stating “As an additional robustness check for multiple comparisons, the comparisons were also run using a Holm-Bonferonni correction, which did not substantially change the results (see Tables S11-S17).”

Only a few comparisons became non-significant, and only one of these was relevant to our main hypotheses. One hypothesis was that trust would drop in the specific source that made an error. Across all Studies we observe this pattern for all sources. A single exception appeared with the Bonferroni-Holm correction in Study 2; the difference between the mainstream error condition and the control condition became non-significant for the DV of mainstream trust (Table S13), which we highlight in the text of the supporting info. Given that Study 2 already showed the weakest pattern of results overall, and that the expected drop in trust remained significant in every other case across 3 studies, we don’t believe that this small adjustment should substantially change the interpretation of our results.

Here are a number of very minor suggestions:

• Line 68 pluralize “vaccine”

• Lines 424-425 “academy awards” should be capitalized as Academy Awards

We have made these corrections, thank you for catching them.

• In Tables, where saying that subscripts apply to the row, perhaps clarify by explicitly saying that they don’t apply between rows

In both the main text and supporting info we have changed the wording to “Within each row, means for each condition that do not share a subscript are significantly different. Subscripts do not compare between rows.”

• Line 724: Under “Trust correlations”, trust in all three sources was not typically positively correlated – they’re sometimes negatively correlated. Please correct how this is described

This is a very good point. We have changed the wording to only indicate positive relationships in Studies 1 and 3, and to point out the negative relationship in Study 2 and add a bit of interpretation: “Notably, in Studies 1 and 3, trust in all three sources were typically positively correlated (with the exception of mainstream and counter-mainstream in Study 1). This is consistent with the General Loss Model, where all sources rise and fall together rather than in competition. It was not the case in Study 2, however, where mainstream and counter-mainstream trust were in fact negatively related (though mainstream and neutral were still positively related). This may reflect the fact that Study 2 examined a real-world issue for which reported trust reflects already-established views. Although we cannot draw strong conclusions based on our correlational results, the finding could indicate that while short-term drops in trust due to a source’s error may be most consistent with the General Loss Model, long-term trends may resemble a more oppositional relationship between the mainstream and counter-mainstream, in line with the Social Alignment Model.”

We have also changed the wording of how we introduce the correlational data in the discussion section (starting on line 802 with tracked changes). The previous short paragraph has been removed, and the opening has been changed to: “Our correlational findings, however, indicate that the immediate effects of our experimental manipulation may only be part of a larger picture. While trust in all of our primary sources were typically positively correlated in Studies 1 and 3, trust in the mainstream and counter-mainstream sources were negatively related in Study 2, which was the only study to examine a real-world, ongoing issue (COVID-19). As well, seeing a source as more politicized was positively associated with trusting its competitors…”

• Table 11 Study 3: Presumably “Counter” means “Counter-Mainstream”. Can this be spelled out in full?

This was originally shortened for formatting reasons, but we have now made it fit in the table with Counter-Mainstream fully spelled-out.

• Line 776: Missing “differed”?

We have added “differed” where it was missing.

• Table 13: in the rows with Source, perhaps specify Mainstream “source”, Counter-Mainstream “source”, and Neutral “source” (i.e., add “source” to the text to help readers figure out quicker what’s varying in rows vs. in columns)

This is a good idea to improve clarity. We have made this change in Table 13 as well as Tables 3, 6, and 7.

Once again, we thank you for your time and effort and for the opportunity to be considered for publication at PLOS One.

---

## [Editor Report · Decision Letter 2]

11 Mar 2025

Is trust a zero-sum game? What happens when institutional sources get it wrong

PONE-D-24-33343R2

Dear Dr. Dawson,

We’re pleased to inform you that your manuscript has been judged scientifically suitable for publication and will be formally accepted for publication once it meets all outstanding technical requirements.

I have two very minor comments, one copyediting, and the other so that the results don’t get undersold. First, under “Trust Correlations”, the 5^th^ sentence appears twice (“It was not the case in Study 2…”, lines 641-644), please delete the second occurrence. Second, I’d recommend adding a short note after you discuss the one finding that became non-significant after the Bonferroni correction (p23 Supplementary), to ensure that some readers don’t reject the finding by saying “well it’s non-significant now!” I’d recommend adding the following at the end of the new paragraph on Supplementary p23: “However, this drop in significance is likely an overcorrection – this finding was predicted in advance, significant in Study 2 without the Holm-Bonferroni correction, and significant in Studies 1 and 3 even with Holm-Bonferroni correction, and as such is a robust and replicated finding.” Such a sentence would reiterate all the strengths, so that people don’t read too much into the fact that the Holm-Bonferroni makes it no longer significant – it’s worth stressing that your results are fairly robust and you replicated them. I’ll leave this to the authors though if they prefer to include or omit such a sentence, but I think it strengthens the manuscript. These changes are so minor that they should not hold up publication of the manuscript – they can be added at the next stage. As such, I’m happy to accept the manuscript at this stage.

Kind regards,

Pat Barclay

Academic Editor

PLOS ONE

Additional Editor Comments (optional):

Thank you for making those changes. I am now happy to accept this manuscript for publication.

I have two very minor comments, one copyediting, and the other so that the results don’t get undersold. First, under “Trust Correlations”, the 5th sentence appears twice (“It was not the case in Study 2…”, lines 641-644), please delete the second occurrence. Second, I’d recommend adding a short note after you discuss the one finding that became non-significant after the Bonferroni correction (p23 Supplementary), to ensure that some readers don’t reject the finding by saying “well it’s non-significant now!” I’d recommend adding the following at the end of the new paragraph on Supplementary p23: “However, this drop in significance is likely an overcorrection – this finding was predicted in advance, significant in Study 2 without the Holm-Bonferroni correction, and significant in Studies 1 and 3 even with Holm-Bonferroni correction, and as such is a robust and replicated finding.” Such a sentence would reiterate all the strengths, so that people don’t read too much into the fact that the Holm-Bonferroni makes it no longer significant – it’s worth stressing that your results are fairly robust and you replicated them. I’ll leave this to the authors though if they prefer to omit such a sentence. These changes are so minor that they should not hold up publication of the manuscript – they can be added at the next stage. As such, I’m happy to accept the manuscript at this stage.
---

## [Editor Report · Acceptance letter]

PONE-D-24-33343R2

PLOS ONE

Dear Dr. Dawson,

I'm pleased to inform you that your manuscript has been deemed suitable for publication in PLOS ONE. Congratulations! Your manuscript is now being handed over to our production team.

Kind regards,

on behalf of

Dr. Pat Barclay

Academic Editor

PLOS ONE